# Data imputation in in situ measured particle size distributions by means of neural networks

Pak Lun Fung[1,2], Martha Arbayani Zaidan[1,2,3], Ola Surakhi[4], Sasu Tarkoma[5], Tuukka Petäjä[1,3] and Tareq Hussein[1,6]

[1]Institute for Atmospheric and Earth System Research / Physics, Faculty of Science, University of Helsinki, Finland; pak.fung@helsinki.fi; martha.zaidan@helsinki.fi; tuukka.petaja@helsinki.fi; tareq.hussein@helsinki.fi

[2] Helsinki Institute of Sustainability Science, Faculty of Science, University of Helsinki, Finland

[3] Joint International Research Laboratory of Atmospheric and Earth System Sciences, School of Atmospheric Sciences, Nanjing University, Nanjing 210023, China

[4] Department of Computer Science, The University of Jordan, Amman 11942, Jordan; ola.surakhi@gmail.com

[5] Department of Computer Science, Faculty of Science, University of Helsinki, Finland; sasu.tarkoma@helsinki.fi

[6] Department of Physics, The University of Jordan, Amman 11942, Jordan

*Correspondence to*: Pak Lun Fung and Tareq Hussein

**Abstract.**

In air quality research, often only size-integrated particle mass concentrations as indicators of aerosol particles are considered. However, the mass concentrations do not provide sufficient information to convey the full story of fractionated size distribution, in which the particles of different diameters ($D_p$) are able to deposit differently on respiratory system and cause various harm. Aerosol size distribution measurements rely on a variety of techniques to classify the aerosol size and measure the size distribution. From the raw data the ambient size distribution is determined utilising a suite of inversion algorithms. However, the inversion problem is quite often ill-posed and challenging to solve. Due to the instrumental insufficiency and inversion limitations, imputation methods for fractionated particle size distribution are of great significance to fill the missing gaps or negative values. The study at hand involves a merged particle size distribution, from a scanning mobility particle sizer (NanoSMPS) and an optical particle sizer (OPS) covering the aerosol size distributions from 0.01 to 0.42 μm (electrical mobility equivalent size) and 0.3 μm to 10 μm (optical equivalent size) and meteorological parameters collected at an urban background region in Amman, Jordan in the period of 1 Aug 2016– 31 July 2017. We develop and evaluate feed-forward neural network (FFNN) approaches to estimate number concentrations at particular size bin with (1) meteorological parameters, (2) number concentration at other size bins, and (3) both of the above as input variables. Two layers with 10–15 neurons are found to be the optimal option. Worse performance is observed at the lower edge ($0.01 < D_p < 0.02$ μm), the mid-range region ($0.15 < D_p < 0.5$ μm) and the upper edge ($6 < D_p < 10$ μm). For the edges at both ends, the number of neighbouring size bins is limited and the detection efficiency by the corresponding instruments is lower compared to the other size bins. A distinct performance drop over the overlapping mid-range region is due to the deficiency of a merging algorithm. Another plausible reason for the poorer performance for finer particles is that they are more effectively removed from the atmosphere compared to the coarser particles so that the relationships between the input variables and the small particles is more dynamic. An observable overestimation is also found in early morning for ultrafine particles followed by a distinct underestimation before midday. In the winter, due to a possible sensor drift and interference artefacts, the estimation performance is not as good as the other seasons. The FFNN approach by meteorological parameters using 5-min data ($R^2 = 0.22–0.58$) shows poorer results than data with longer time resolution ($R^2 = 0.66–0.77$). The FFNN approach by the number concentration at the other size bins can serve as an alternative way to replace negative numbers in size distribution raw dataset thanks to its high accuracy

and reliability ($R^2 = 0.97$–1). This negative numbers filling approach can maintain a symmetric distribution of errors and

complement the existing ill-posed built-in algorithm in particle sizer instruments.

**Keywords.**

Atmospheric aerosols particles, feed-forward neural network, interpolation, missing data, SMPS, OPS

## 1 Introduction

Particulate matter (PM) is the principal component of air pollution. PM includes a range of particle sizes, such as coarse

(1< particle diameter ($D_p$)<10 µm), fine (0.1< $D_p$<1 µm), and ultrafine particles (UFP, $D_p$< 0.1 µm). Through human's

inhalation, coarse particles usually are partly deposited in the head airway (5–30 µm) by the inertial impaction mechanism,

and are partly deposited in the tracheobronchial region, mainly through sedimentation (1–5 µm). The particles may be

further absorbed or removed by mucociliary clearance (Gupta and Xie, 2018). The remaining fine and UFP, due to their

high surface area to mass ratios (Kreyling et al., 2004), penetrate deeply into the alveolar region, where removal

mechanisms may be insufficient (Gupta and Xie, 2018). Evidence suggests that the adverse associations of short-term

UFP exposure with acute and chronic problems ranging from inflammation, exacerbation of asthma, and metal fume fever

to fibrosis, chronic inflammatory lung diseases, and carcinogenesis (Spinazzè et al., 2017) might be at least partly

independent of other pollutants (Ohlwein et al., 2019). Various studies have demonstrated that inhaled or injected UPF

could enter systemic circulation and migrate to different organs and tissues (Londahl et al., 2014; Xing et al., 2016).

Other than health effects, particles of various sizes also contribute to Earth's ecosystem and climate differently. For

instance, fine and UFP are capable of growing up to diameters of 0.02–0.1 µm within a day (Kulmala et al., 2004;

Kerminen et al., 2018) where they constitute a fraction of cloud condensation nuclei; thus, indirectly affecting the climate

(Kerminen et al., 2012). The drivers behind aerosol particles vary between natural and anthropogenic as well as primary

and secondary. Primary particles are emitted to the atmosphere as particles, such as sea salt or dust particles, while

secondary particles form in the atmosphere through gas-to-particle transformation, which has been known as new particle

formation (NPF) observed in various environments and contributing to a major fraction of the total particle number budget

(Kulmala et al., 2004; Kerminen et al., 2018). In addition, while fine particles cool the climate by predominantly scattering

shortwave radiation, coarse particles warm the climate system by absorbing both shortwave and longwave radiation (Kok

et al., 2017). Indeed, the complexity of urban aerosols is tribute to the fact that several sources can contribute in the same

particle size range (Rönkkö et al., 2017).

Currently, the most commonly reported aerosol variables are particle mass concentration and particle number

concentration. The former metric, which is dominated by coarser particles, is included as air quality indicators (e.g. mass

concentrations of both thoracic particles $PM_{10}$ and fine particles $PM_{2.5}$); however, it has been argued that this might ignore

the potential adverse effect of UFP on health (Zhou et al., 2020). The latter one describes better the distribution of finer

particles, but it neglects the influence of coarse particles. Using either particle mass concentration or particle number

concentration solely is not enough to fully review the health effects and the Earth's climate system by aerosol particles.

Therefore, in order to understand the origin of atmospheric aerosol particles and their potential impacts at a specific

location, the whole size distribution of these particles needs to be studied (Zhou et al., 2020).

Recently, due to urbanization and increased population, megacities have increased their contribution to atmospheric aerosol pollution massively Lelieveld et al. (2015). Middle East and North Africa (MENA) regions, with an average annual growth rate of 1.74% in 2019 (World Bank Group, 2019), has one of the world's regions most rapidly expanding populations. With the population of 578 million, several cities in MENA regions are among the 20 most polluted cities in the world. The annual average concentrations of some pollutants, for example PM$_{2.5}$ in MENA (54.0 μg m$^{-3}$) often exceed 5 times the WHO recommended levels (10.0 μg m$^{-3}$) (World Health Organisation, 2019). Many countries in MENA are dealing with negative impacts of air pollution in terms of both economic burden and health aspect (Ahmed et al., 2017; Goudarzi et al., 2019). Air Pollution in this region is estimated to cause 133,000 premature deaths annually, almost half of which are attributed to natural sources of air pollution, such as windblown sea salt and desert dust (Gherboudj et al., 2017). Apart from natural pollutants, anthropogenic activities also play a major role in driving the air quality. They include the extensive development of petrochemical industry, vehicular emissions and open burning of waste (Arhami et al., 2018).

However, aerosol studies in this region have not paid attention to the aerosol number size distribution so far. Among the few studies published, most report mass concentration (Goudarzi et al., 2019; Arhami et al., 2018; Borgie et al., 2016), while some focused on the total particle number in MENA regions. Studies on the size-fractionated number concentrations are, nonetheless, scarce (e.g. Hakala et al., 2019) due to the unavailability of instruments for measuring UFP in many air quality monitoring stations (Spinazzè et al., 2017). Determining aerosol number size distribution for a wide size range in a reliable manner is a challenging task. The fact that the ambient distributions range from nanometers to several micrometers dictates the use of multiple sizing techniques. For the sub-micron size range, electrical mobility equivalent diameter is commonly used as the size parameter and the measurements are performed with Differential Mobility Particle Sizer (DMPS) or Scanning Mobility Particle Sizer (SMPS) instruments (e.g. Wiedensohler et al., 2012) . These systems determine the aerosol size according to electrical mobility equivalent size. The larger particles (approximately > 0.3 μm) can be classified according to their aerodynamic or optical size (Kulkarni et al., 2011). In order to obtain the full aerosol size distribution, this data needs to be merged. Unfortunately this task is not trivial as the merging requires knowledge on the chemical composition (influencing the refractive index and thus the optical size), shape (influencing electrical mobility equivalent size), or effective density (influencing aerodynamic size) (Kannosto et al., 2008).

In addition, the raw data from these instruments must be inverted to obtain the particle size distribution. This is not a straightforward problem. A proper inversion algorithm is required to restore the particle size distribution from the raw response (Cai et al., 2018) using recorded kernel functions which describe the probability of particles of a certain size being measured at a certain flow rate, influenced by the measured activation curves and the detection efficiencies of the instruments (Lehtipalo et al., 2014). Depending on the instruments used and the measurement environments, some use a built-in inversion algorithm in the instruments, which replace negative raw values with artificial non-negative numbers. Some develop their own inversion methods; however, they all have their drawbacks. Examples include that the least square method may magnify the random errors in the raw counts in Condensation Particle Counter (CPC) into relatively large uncertainties (Enting and Newsam, 1990), the stepwise method may cause non-negligible errors (Lehtipalo et al., 2014), and that the smoothing step method may introduce bias in the shape of the inverted distribution function (Markowski, 1987). Kandlikar and Ramachandran (1999) pointed out that there is not a single universal inversion algorithm applicable to all situations. In this study, the built-in inversion algorithm was used. This algorithm can lead to negative values when the kernel functions are not optimally configured, especially in the size range of low number

concentration. These negative values have no physical meanings. Some experts in the in situ measurement community might just omit the negative values or simply use nearest neighbour linear interpolation to replace the negative values. However, the former method might cause asymmetric error for very small measured number concentration values (Viskari et al., 2012), while the latter could result in too high values concurrently. To fill this knowledge gap, statistical estimation methods can serve as an alternative to estimate of size-fractioned number concentration by using other available measurements.

The main objective of the paper is to estimate particle number concentration/ fill the negative values making up for the shortcomings of the built-in inversion algorithm in particle sizer instruments. Extending from the previous study by Zaidan et al. (2020), we build our imputation method with a finer temporal and size-bin resolution. In order to do so, we place emphasis on estimating particle number concentration of a specific size bin by the interaction with other size bins and meteorological variables. In this study, we propose three approaches in terms of different input variables by means of neural networks: (1) only meteorological parameters, (2) only particle size distribution, and (3) both particle size distribution and meteorological parameters. Based on the general data analysis of the particle size distribution and the meteorological condition, we explain the source of different size bins at certain weather conditions and the correlation among the particle size distribution and meteorological parameters in Sect. 3. We evaluate the proposed neural network method and compare it with other simpler methods in Sect. 4.1. In Sect. 4.2, we further discuss the temporal pattern of the proposed method in terms of its diurnal cycle, weekend effect and seasonal variation. Besides, we examine the possible technical reasons for the pattern found and the application of the proposed method.

## 2 Methods

### 2.1 Measurement sites and Instruments

In this study, we collected a dataset obtained from a measurement campaign in Amman, the capital city of Jordan, between 1 August 2016 and 31 July 2017. The city represents an area with Middle Eastern urban conditions within the Middle East and North Africa (MENA) region. This region serves as a compilation of different aerosol particle sources including natural dust, anthropogenic pollution (e.g. generated from the petrochemical industry and urbanization), as well as new particle formation.

The database includes particle size distribution and meteorological parameters, as mentioned in the first step in Figure 1. The aerosol measurement was carried out at the aerosol laboratory located on the third floor of the Department of Physics, University of Jordan (32°00′ N, 35°52′ E) in the neighbourhood of Al Jubeiha. The campus is situated at an urban background region in northern Amman. In particular, the campaign measured the particle number size distribution using a scanning mobility particle sizer (NanoScan SMPS 3910, TSI, MN, USA) with default settings. It monitors the particle size distributions as electrical equivalent diameter 0.01–0.42 μm (13 channels). The size range of the SMPS system can be extended to coarse particles with an additional compact instrument: an optical particle sizer (OPS 3330, TSI, MN, USA). OPS measures optical diameter 0.3–10 μm (13 channels). This optical sizing method reports an optical particle diameter, which is often different from the electrical mobility diameter measured by the SMPS technique. The measurements were combined to provide a particle size distribution of wider particle diameter range 0.01–10 μm, which is further described in Sect. 2.2. The SMPS inlet consists of copper tubing with a diffusion drier (TSI 3062-NC). The inlet

flow rate was 0.75 lpm (±20%) while the sample flow rate was 0.25 lpm (±10%). The flow rate of OPS was about 1 lpm. The aerosol transport efficiency and losses through the aerosol inlet assembly and the diffusion drier was determined experimentally in the laboratory: ambient aerosol sampling alternatively with and without sampling inlet, and the aerosol data was corrected accordingly. The penetration efficiency was ~47% for 0.01 μm, ~93% for 0.3 μm and ~40% for 10 μm (Hussein et al., 2020). These deficiency of measurement at the upper and lower edges is somewhat in alignment with other literatures. Particle size measured by nanoSMPS (Tritscher et al., 2013) tended to be underestimated for spherical particles larger than 0.2 μm by up to 34% (Fonseca et al., 2016). Liu et al. (2014) clearly portrayed that the detection limit of particle size below 0.03 μm is about 80–500 $cm^{-3}$, which is up to 10 times larger than that of coarser particles, for other versions of SMPS. Stolzenburg and McMurry (2018) explained that discrepancies could be resulted from Differential Mobility Analysers (DMAs) with transfer functions that were degraded (i.e., broadened) by flow distortions caused by particle deposition within the classifier tube, sizing errors due to errors in flowmeter calibrations or leaks, CPC concentration errors due to improper pulse counting, and continuity failure in the DMA high voltage connection.

The meteorological measurement was performed with a weather station (WH-1080, Clas Ohlson: Art.no.36-3242, Helsinki, Finland) with a time resolution of 5 minutes. The meteorological data were comprised of ambient temperature (Temp, resolution 0.1°C), relative humidity (RH, resolution 1%), wind speed (WS), wind direction (WD, 16 equal divisions) and air pressure (P, resolution 0.3 hPa) (Hussein et al., 2019; Hussein et al., 2020; Zaidan et al., 2020). Wind direction is resolved into north and east direction, as WD-N and WD-E, respectively. The data collection process is illustrated in the first step in the database block in Figure 1.

**2.2 Data pre-processing**

The next step in the database block in Figure 1 is data pre-processing. Since the sampling time resolution of SMPS and OPS was 1 min and 5 min, respectively, we synchronised the data into 5-min averages. Since a part of the size ranges in both instruments are overlapping with each other, the last two size bins in SMPS and the first size bin in OPS were neglected. Finally, we merged the size range of electrical mobility diameter 0.01–0.25 μm by SMPS and optical diameter 0.32–10 μm by OPS, and obtain a wider particle size distribution which covers the diameter range 0.01–10 μm. Merging electrical mobility diameter and optical diameter can be a challenge and the overlapping region is often calculated with high uncertainty (DeCarlo et al., 2004; Tritscher et al., 2015). The challenge arises because the optical diameters are measured based on the refractive index of the particles, which depends on their chemical composition. Therefore, the sizing will vary over time. There is also a slight dependency with the SMPS system that is linked to the shape of the particles, which influences their sizing.

We also calculated the particle number concentration with four particle diameter modes (size-fractionated number concentration): nucleation (0.01–0.025 μm), Aitken (0.025–0.1 μm), accumulation (0.1–1 μm) and coarse mode (1–10 μm). Subsequently, the total number concentration was obtained as the sum of all these fractions. The size-fractionated number concentrations were obtained by summing up the measured particle number size distribution over the specified particle diameter range.

In order to perform data imputation with neural networks, aerosol and meteorological data were first linearly interpolated in time in case of short missing data periods. For missing data over longer periods, the whole rows are eliminated. The

196 shorter missing data occurs due to technical faults while the longer missing periods are attributed to instrument
maintenance (Zaidan et al., 2020). Only 71.8% of total data was retained for the next step in the measurement period.
Since the data were obtained from different measured variables with various physical units and magnitudes, it was crucial
to normalise the data. The scaling factor depends on which activation function is chosen. In this case, the datasets were
scaled so that it has a mean of 0 and a standard deviation of 1 to transform them into the range of the activation function.
The standardised data was then separated into different months for the reason of the seasonal variation in the atmospheric
condition. The data was further divided into training set (70%) and testing set (30%). The processed data were also
converted to hourly and daily averages for reporting purposes.

**2.3 Setting of the neural network**

After data collection and data pre-processing procedures, the next step is method optimisation (Figure 1). ANN models
have been utilised in predicting air quality (Freeman et al., 2018; Maleki et al., 2019; Cabaneros et al., 2019; Zaidan et
al., 2020; Fung et al., 2020). Neural networks provide a robust approach for approximating real-valued target functions
because they can mimic the non-linearity of the functions and their optimisation methods are well developed (Zaidan et
al., 2017). The architecture of neural networks consists of nodes as activation function (Figure 2), and the activation
function in each layer determines the output value of each neuron that becomes the input values for neurons in the next
hidden layer connected to it. In this paper, feed-forward neural network (FFNN) is used instead of a more sophisticated
time delay neural network (TDNN) because some of the rows in the dataset were removed in the data pre-processing step
due to the existence of missing data and TDNN cannot be performed without time continuity. FFNN usually consists of
a series of layers. The first layer has a connection from the network input. Each subsequent layer has a connection from
the previous layer. The final layer produces the network's output. A neuron can be thought as a combination of two parts:

$$z_j^{(L)} = \sigma\left(\sum_{i=1}^{n} w_{ji}^{(L)} x_i + b_j^{(L)}\right)$$

(1),

where $z_j^{(L)}$ and $b_j^{(L)}$ are the intermediate output and the bias term for the $j^{th}$ neuron at $L^{th}$ layer, respectively. $w_{ji}^{(L)}$ is the $j^{th}$
weight for each data points $x_i$ at $L^{th}$ layer. The second part performs the activation function (sigmoid function in this
study) on $z_j$ to give out the output of the neuron:

$$\sigma\left(z_j^{(L)}\right) = \frac{1}{1 + \exp^{-z_j^{(L)}}}$$

(2),

The FFNN method was created, trained and simulated with MATLAB (version: 8.3.0.532), using Neural Network
Toolbox. We initialised the weights randomly and the weights were updated through ''Levenberg-Marquardt'' algorithm
optimisation that was the fastest available back-propagation training function (Chaloulakou et al., 2003). We performed
several iterations within a cycle to minimise the training loss with Bayesian regularisation. These steps were done
iteratively until the best combination of the number of hidden layers and the corresponding number of neurons that
provided the minimum error was found. According to the review paper by Cabaneros et al. (2019), a shallow neural
network with one hidden layer and enough neurons in the hidden layers can fit any finite input-output mapping problem
for non-linear relationship. In the network training process, the number of neurons varied from 2 to 10 neurons per layer
with an incremental factor of 2 neurons in each simulation, and from 10 to 25 per layer with an incremental factor of 5
neurons in each simulation. To keep the method simple, we consider only one or two layers in the simulation process
because the computing requirements could rise exponentially with the number of layers and neurons. Once we pick the
suitable method configuration, the method estimates number concentration using testing data. Finally, the selected

performance metrics, described in Section 2.4, can be calculated and we evaluate which approach is the most suitable for size distribution estimation.

**2.4 Other methods as comparison with the neural network method**

In order to demonstrate the performance of the FFNN method, we perform similar procedures applying other simpler methods, which have been widely used as means of data imputation (Junger and Ponce De Leon, 2015). They include univariate and multivariate methods. The former includes unconditional mean (UM), median (MD), linear interpolation (LinI), logarithmic interpolation (LogI), next neighbour interpolation (nNI) and previous neighbour interpolation (pNI), where nNI was implemented as the next value carried backward while pNI as the previous value carried forward. The multivariate methods used in this study are conditional mean based on a linear regression of meteorological parameters and other particle size number concentrations as inputs (CM–met and CM–PSD, respectively). These methods are implemented as a comparison with the FFNN method.

**2.5 Performance metrics**

We choose the optimal combination of the number of hidden layers and the corresponding number of neurons by checking its mean absolute error (MAE), which is a simple way to illustrate the residuals of the estimated values by the estimation method. In order to identify which size bin manage to be predicted best, two metrics are used, namely coefficient of determination ($R^2$) and normalised root-mean-square error (NRMSE). $R^2$ measures how well the observed outcomes are replicated by the estimation method, based on the proportion of total variation of outcomes explained by the estimation method. NRMSE represents the standard deviation of the estimated errors with respect to its mean. NRMSE is used rather than commonly used RMSE because the number concentrations of the different size range are of different magnitudes. The comparison in different size range becomes different if RMSE is not normalised with its mean.

$$\text{MAE} = \frac{\sum_{i=1}^{n} |y_i - \hat{y}_i|}{n} \tag{3}$$

$$R^2 = 1 - \frac{\sum_{i=1}^{n}(y_i - \hat{y}_i)^2}{\sum_{i=1}^{n}(y_i - \bar{y})^2} \tag{4}$$

$$\text{NRMSE} = \frac{\sqrt{\frac{\sum_{i=1}^{n}(y_i - \hat{y}_i)^2}{n}}}{\bar{y}} \tag{5}$$

where $y_i$, $\hat{y}_i$ and $\bar{y}$ represent the $i$th measurement value, the $y$th estimated value by the estimation method and the mean of the all the measurement data, respectively. $n$ notates the total number of the valid measurement data.

**3 General data analysis**

**3.1 Environmental condition**

Hussein et al. (2019) and Zaidan et al. (2020) investigated and described the effect of local weather conditions, respectively. Here we describe briefly the meteorological conditions during the measurement period as background information. Starting from August 2016, the daily temperature decreased gradually from 40°C to its tough 0°C in February 2017. It rose gradually to 40°C in August 2017. During the measurement period, the hourly median value was 19.9°C (Figure 3a). RH varied quite a lot from 10% to 100%, with an hourly median of 52.3%, and did not seem to have a seasonal pattern (Figure 3b). In summer months, wind appeared be stronger but the wind direction is more stable, mostly

from northwest (270º–360º). In cold months, averaged wind speed was lower but wind blew from fluctuating direction.

During the whole measurement period, wind speed ranged between 0–6 m s$^{-1}$ and its median is 1.39 m s$^{-1}$ (Figure 3c–d).

Air pressure varied in a range from 892 to 912 hPa and its hourly median was 900 hPa In spite of the narrow range of

variation, winter months seem to have slightly higher air pressure than summer months (Figure 3e).

Meteorological conditions have been suggested to influence particle number concentration. Hussein et al. (2019)

demonstrated that number concentration had a rather complex relationship with temperature. Furthermore, number

concentration of submicron had a decreasing trend with respect to the wind speed which indicates that most of the

submicron fraction is originated from local sources such as combustion processes. Meanwhile, the number concentration

of coarse particles had higher concentrations at stagnant conditions and when the wind speed is higher than 5.5 m s$^{-1}$. It

is mainly because of road dust resuspension and might also be attributed to dust storm via long-range transport Hussein

et al., 2019. In this study, we further explore how wind direction influences the particle number concentration (Figure 4).

Wind coming from the northwest (225º–325º) was generally stronger, but lower particle number concentration was

detected because the measurement area is at the outskirt of downtown. Wind from East and South (45º–225º) has a lower

wind speed but a more intense hourly particle number concentration can be detected. From that direction situates the

urban city where all kinds of industrial activities take place. When considering only coarse particles, relatively high

number concentration is found when south-westerly wind is strong. This can further serve as an evidence that the source

of coarse particles in that region might come mostly from long range sea salt from Dead Sea or dust particles from nearby

deserts.

**3.2 General pattern of particle size distribution**

Hourly total number concentration ranged from $1.90 \times 10^3$ cm$^{-3}$ to $1.52 \times 10^5$ cm$^{-3}$ and its median was $1.36 \times 10^4$ cm$^{-3}$. Figure

5a performed moderate seasonal pattern in general: lower in summer months and higher in colder months. Hussein et al.

(2019) also characterised the modal structure of the particle number size distribution for the same site. Four modes have

been detected by lognormal fitting, as known as DO-FIT algorithm and modal structure (Hussein et al., 2005; Hussein et

al., 2019), revealed that the mode number concentrations of the nucleation, Aitken, and coarse modes were lognormally

distributed around their geometric mean values: 0.022 μm, 0.062 μm, and 2.3 μm respectively. However, the accumulation

mode number concentration had two distinguished modes with particle diameter centred at 0.017 μm and 0.39 μm. As

seen in Table 1, the total number concentration of all particle size ($1.70 \pm 1.26 \times 10^4$ cm$^{-3}$) is mostly accounted by Aitken

mode (45–80%, average: $1.09 \pm 1.01 \times 10^4$ cm$^{-3}$), followed by nucleation mode (10–50%, average: $0.48 \pm 0.32 \times 10^4$ cm$^{-3}$).

Accumulation mode (0–15%, average: $0.13 \pm 0.08$ cm$^{-3}$) comes third and only less than 0.5% of the total particle number

concentration contain coarse particles with an average of $2.13 \pm 2.80$ cm$^{-3}$ (Figure 5b–e). Seasonal pattern of the total

number concentration resembles the Aitken composition: lower proportion in summer months and higher in colder

293  months. The ratio of nucleation mode performs in an opposite way. The seasonal variation of total number concentration

is due to the more suppressed boundary layer in winter (Teinilä et al., 2019) and the elevated wood combustion (Hellén

et al., 2017). The particle number of accumulation and coarse mode steadily stay at a low proportion line, which did not

account for the total number concentration. It is also noticed that dust episodes occurred with the concentrations that often

exceeded 2 cm$^{-3}$ and the daily concentration in the course of these episodes can rise to 20 cm$^{-3}$. These episodes were often

found in spring from February to May and some episodes can last for up to one week.

Similar to many other urban environments, the diurnal pattern observed in this study reflects the combustion emissions from traffic activity, which is more during the workdays (Hussein et al., 2019). The two peaks of the nucleation mode and Aitken mode in the cold months are relevant for the morning and the afternoon traffic rush hours, which are similar to those noticed in most cities in other countries. In warmer months, the diurnal cycles are not as distinct, but a sharp peak of nucleation mode around noon is found, which is associated with the occurrence of new particle formation. These events occurred very often in the summer as suggested by Hussein et al. (2020). The amplitude of diurnal cycles of coarse mode is small while the patterns of accumulation are not clear (Figure 6).

**3.3 Correlation analysis**

Figure 7 demonstrated the interaction among the whole measured spectrum shows three range clusters based on their correlation with the number concentration at other bin sizes: 0.01–0.205 μm, 0.205–0.875 μm and 0.875–10 μm. 0.01–0.205 μm and 0.875–10 μm fall entirely within the size range detected by SMPS and OPS, respectively. The 5-min number concentration of smaller size and bigger size bins have clear and strong correlation with the concentration of its neighbouring size bin. However, particles of size 0.205–0.875 μm are located in the overlapping regions by the two instruments; as a result, do not correlate well with other size bins. The correlation of 5-min particle size distribution with meteorological parameters are generally low. Temperature appears to be the most correlated parameters for all bin sizes among all the parameters we used in this study. Smaller size range have higher Pearson's correlation coefficient (R) than larger size range for WD, WS and P.

The 5-min averaged data show similar correlation for the particle size distribution except for the smallest size bin. The hourly and daily data have higher correlation with the other size bins which are also monitored by SMPS. The 5-min averaged data show different correlation from the hourly and daily averaged data performed by Zaidan et al. (2020). The correlations of 5-min size distribution with all meteorological variables are below 0.5 for all size range. However, for hourly and daily averaged data, R is much higher in specific size bins. Hourly and daily temperature, in particular, show increasing R with larger particle size for accumulation and coarse mode. Overall, the correlations increase with the longer averaging windows. This might be due to the buffer period the meteorological conditions act on the dispersion of particles. Based on this result, using data with finer temporal resolution might be considered to be less influential to the estimation accuracy.

**4 Evaluation of the proposed method**

**4.1 General evaluation**

Figure 8 illustrates how well the three approaches of the proposed FFNN perform in term of $R^2$ and NRMSE.

**Approach 1 (Size distribution estimation based on meteorological parameters only, FFNN–met):** For more than half out of the 23 size bins, 2 layers and 15 neurons is the best combination where the residuals are the lowest (Table 2). Owing to the poor correlation with meteorological condition, we expect a low correlation of determination even using the optimal configuration neural network ($R^2 = 0.22$–$0.58$). The $R^2$ are low at the nucleation mode ($0.01 < D_p < 0.03$ μm) of the whole size distribution around nucleation mode ($R^2 \sim 0.2$). The rest of the size bins have better and more stable performance ($R^2 = 0.4$–$0.58$). This shows that the instrument might have a poor detection efficiency for particles of smaller size. The performance of FFNN method using 5-min data for all size bins ($R^2 = 0.22$–$0.58$) is worse than using daily data

($R^2 = 0.77$) performed in Zaidan et al. (2020). Compared with hourly data ($R^2 = 0.66$), the overall performance of the

method using 5-min data is comparable ($R^2 = 0.67$).

**Approach 2 (Size distribution estimation based on other particle sections only, FFNN–PSD):** This approach works

well with most combination of number of layers and neurons. They do not show a clear difference among the combinations

we choose. There is no single combination which entirely outperform the others in all size bins. We summed up the MAE

for all size bins and decided to stick to 2 layers and 10 neurons with the overall lowest residuals (Table 2). $R^2$ are all

above 0.97 for all bin sizes, and NRMSE are 0.01–0.25 for all bin sizes. The results are expected because there are 22

inputs and one output. Relatively worse correlation at the edges of size bins ($0.01 < D_p < 0.02$ μm; $6 < D_p < 10$ μm) is found

because of the lack of nearby size bins which has high correlation with the corresponding size bin. Another reason could

be that the instrument has a higher detection limits for smaller particles (Liu et al., 2014). The poorer performance for

smaller size might be due to a coarser sizer resolution compared to other SMPS components (Tritscher et al., 2013), so

that NanoSMPS does not reflect the real enough size distribution in the atmosphere. Relatively poor estimation

performance at the middle size range ($0.15 < D_p < 0.5$ μm) in the whole measured spectrum is because of the overlapping

of instruments. This also ascertain the importance of creating a better algorithm when we merge two or more size

distribution by different instruments. In this study, the measuring techniques and the measuring targets are different by

the SMPS and OPS. The merging of the two measuring targets, the optical particle diameter and the electrical mobility

diameter, might create significant uncertainties (DeCarlo et al., 2004; Tritscher et al., 2015). The estimation of certain bin

size by other bin sizes can be thought of replacing negative values in the raw data by particle sizers. While some instrument

manufacturers create built-in algorithms to replace with artificial non-negative numbers, most end-users simply remove

the seemingly impossible negative values from the dataset. The perfect way to do it is to have a parallel instrument that

overlaps with that particle size range. However, in many cases, this is not possible as a result of financial constraints.

Therefore, we shall rely on the mutual relationship between the size sections in the aerosol population. Negative values

appear often at size bins with very low number concentration (usually in coarse mode). Instead of eliminating them, this

alternative could maintain the symmetry of the error distribution of the number concentration (Viskari et al., 2012) and

minimise the uncertainties caused.

**Approach 3 (Size distribution estimation based on meteorological parameters and other particle sections):** The

general results are similar as in PSD. However, the more input variables do not enable the approach to work better. At

some bin size the $R^2$ are even slightly smaller than PSD solely. Since meteorological data show low correlation with most

portion of measured spectrum. In that approach, the addition of meteorological parameters is not beneficial to the

estimation process. Due to the lack of improvement in the method development, we will only focus on the two methods:

FFNN–met and FFNN–PSD from now on.

In order to highlight the performance of the FFNN methods in terms of accuracy and reliability, we compare the FFNN

methods with other simpler methods, the results as shown in Table 3 for $R^2$ and Table 4 for NRMSE. The $R^2$ of the

univariate methods UM and MD are close to 0 because their imputation are over-simplified and imply the replacement of

a missing value by a constant. This can be further validated by the narrow range of the estimated particle concentrations

in Figure 9a–b. The remaining univariate interpolation methods LinI, LogI, nNI and pNI showed good results in general

($R^2 = 0.82$–0.92, NRMSE = 0.57–0.88), but failed to perform even fairly at some particle size bins. This implies that these

methods are not stable for the whole spectrum of the particle size distribution. Some of the estimated particle

concentrations are off from the 1:1 line, which implies that the estimation of some particle bins are not as accurate (Figure

9c–f). The performance results of the multivariate methods CM–met and CM–PSD are comparable to FFNN–met and FFNN–PSD, but both CM methods show weaker performance than FFNN methods in terms of $R^2$ and NRMSE no matter whether meteorological (CM–met: $R^2 = 0.52$, NRMSE = 1.39; FFNN–met: $R^2 = 0.67$, NRMSE = 1.13) or particle size distribution data (CM–PSD: $R^2 = 0.99$, NRMSE = 0.17; FFNN–PSD: $R^2 = 1.00$, NRMSE = 0.07) is used as inputs. The pattern of performance of the multivariate methods is also similar to those of FFNN, i.e., relatively poor performance at the edges of size bins ($0.01 < D_p < 0.02$ μm; $6 < D_p < 10$ μm) and the overlapping region ($0.15 < D_p < 0.5$ μm). When combining the whole spectrum, FFNN methods (Figure 9i–j) appear to have narrower bands than CM methods (Figure 9g–h) along 1:1 line, which indicate the methods work similarly across the particle size spectrum. Although the multivariate method CM–PSD (Figure 9h) also rely on the mutual relationship between the size sections in the aerosol population, this method is not as accurate and stable as our proposed FFNN–PSD.

From the perspective of physics, particles in the nucleation mode ($0.01 < D_p < 0.03$ μm) are more sensitive to transformation processes due to their volatility and rather unstable nature (Morawska et al., 2008). This leads to a relatively short lifetime in the atmosphere (Al-Dabbous et al., 2017), hence, the relationships between the input variables and the nucleation mode are not well established. Al-Dabbous et al. (2017) demonstrated that accumulation mode particles ($0.1 < D_p < 0.3$ μm) have much longer lifetimes compared to smaller particles, causing them to be transported for larger distances (Laakso et al., 2003); therefore, the mapping of the relationships between long–range transported accumulation mode particles and covariates is supposed not to well understood. However, the relative prediction ability in this study is not lower given that local meteorological variables were used as input variables. The possible reason is that this mode falls exactly in the instrumental overlapping regions, which leads to a lower predictability. The locally-produced Aitken mode particles ($0.03 < D_p < 0.1$ μm) are less effectively removed by transformation processes (e.g. evaporation and coagulation) from the atmosphere, compared with nucleation mode ($0.01 < D_p < 0.03$ μm), allowing the estimation methods to better understand their relationships with the input variables, which is in alignment with Al-Dabbous et al. (2017).

## 4.2 Temporal pattern

Figure 10 shows the diurnal discrepancies during workdays and weekends. Relative particle number concentration was defined by the estimated concentration with respect to the measured concentration. Values above 1 indicates overestimation while values below 1 suggests underestimation. For approach 1 (FFNN–met), except for the overlapping size bin, which are underestimated by more than 50% at all time range, the difference between estimated and measured hourly number concentration is within 50% during both workdays and weekends. Overestimation is found in early morning before 3 a.m. during workdays for all size bins, especially for UFP. Following the overestimation, at about 6 a.m. in the morning, the estimated number concentration appears to understate by up to 40%, especially at size bins below 0.1 μm. Along the day, the estimation uncertainties are rather small until in the evening from 6 p.m. to 11 p.m. where estimated UFP number concentration show moderate overestimation one more time. It reveals that FFNN–met fails to catch the diurnal pattern from 6 p.m. to 7 a.m. in particular for UFP. The pattern of the performance for weekends does not appear to be as distinctive as on workdays. It shows the overestimation not only for UFP in early morning about 3 a.m., but also at the upper edge larger than 5 μm from 3 a.m. to 4 p.m.. At 7.p.m. onwards until noon, an underestimation is found at all size bins. For approach 2 (FFNN–PSD), except the overlapping size bin, which has a significant overestimation from 6 p.m. to 7 a.m., most show negligible 10% uncertainty during both workdays and weekends. The

performance over weekends show relatively stronger uncertainties. The smallest bin at 0.01 μm is slightly understated for all hours of a day. Other than these, FFNN–PSD manages to catch fairly well the diurnal pattern for all size bins.

Figure 11 further shows the monthly deviation in estimation performance. For approach 1 (FFNN–met), higher $R^2$ is found in November, February and April in the range of SMPS. Other than that, no observable variation in $R^2$ in approach 1 (FFNN–met). For approach 2 (FFNN–PSD), except in January when all the rows were eliminated because of the lack of wind information, performance in the other months is steady for most size range. At 0.21 μm, the difference in estimation performance varies across different months. $R^2$ in winter months are 0.76, 0.36 and 0.61, in November, December and February, respectively, while $R^2$ exceeds 0.9 in other months. This unexpectedly low $R^2$ only occurs in the winter months at the overlapping size range. It can be speculated that the measurements by the two instruments differ in a larger extent during winter. This might be attributed to sensor drift and a number of interference artefacts for particle measurements associated with several factors, such as relative humidity, temperature and other gas-phase species, which were demonstrated by several researchers (e.g. Lewis et al., 2016; Popoola et al., 2016). Another reason for the difference in estimation performance can be that the percentage of complete rows in these months are lower than the other months. The drop in data points might impose an influence to the estimation performance. Especially in June, at the few size bins close to the larger edge, $R^2$ ranges from 0.9 to 0.7. Besides that, some low $R^2$ can be also found in individual month at both edges of size range, which does not appear to show any patterns.

In short, the estimation ability for lower edge ($0.01 < D_p < 0.03$ μm) is found worse in both approaches. The performance of the FFNN method in mid-range ($0.15 < D_p < 0.5$ μm) and upper edge ($6 < D_p < 10$ μm) are relatively worse for the approach with other fractionated size bins as input variables according to the aforementioned statistical performance indicators. All statistical estimation simulations are based on the previous history of relationships between the inputs and outputs. As a result, the estimation simulations for different size ranges have significantly unique connections. The approach by meteorological parameters considers only 6 predictor variables so the accuracy is lower than FFNN–PSD. It might not seem surprising that the deviations between the measured and estimated size distribution were not substantial ($R^2 > 0.97$, NRMSE$< 0.25$) because FFNN–PSD takes 22 other size bins as predictor variables. This, still, gives a clue that the proposed FFNN method can provide adequate solutions to particle size distribution prognostic demands. Furthermore, this FFNN method outperforms the other selected widely used methods in terms of its accuracy and reliability. The estimation of certain bin size by other bin sizes can be thought of replacing 'negative' values in the raw data by particle sizers, including SMPS we used in this paper. Instead of eliminating the negative values, they can be estimated by other size bins with a high accuracy in order to keep the symmetry in data error distribution (Viskari et al., 2012).

**5 Conclusion**

This paper presents the evaluation of imputation methods by means of feed-forward neural network (FFNN) for estimating particle number concentration at various particulate size bins. Input predictors include a merged particle size distribution, by a scanning mobility particle sizer (NanoSMPS) and an optical particle sizer (OPS), which covers size range from 0.01 to 10, and meteorological parameters, including temperature (Temp), relatively humidity (RH), wind speed (WS), wind direction (WD) and ambient pressure (P). The measurements were collected in an urban background region in Amman, the capital of Jordan in the period of 1 Aug 2016–31 July 2017. The total number concentration ($1.70 \pm 1.26 \times 10^4$ cm$^{-3}$) in

the measurement period show moderate seasonal variability owing to the more suppressed boundary layer (Teinilä et al., 2019) and the elevated wood combustion (Hellén et al., 2017) in wintertime. Similar to many other urban environments, the diurnal pattern observed in this study reflects the traffic activity, which has a more pronounced pattern during workdays (Hussein et al., 2019). The amount of coarse particles is negligible in terms of number concentration but dust episodes were found often in spring during the measurement period.

We proposed three approaches with different input variables: (1) only meteorological parameters, (2) only number concentration at the remaining size bins, and (3) both of the above. We performed optimisation to obtain the optimal configuration of the FFNN methods, which are two layers with 10–15 neurons, balancing the accuracy and the computing resources. The 5-min averaged meteorological parameters give varying number concentration estimation for various size bins ($R^2$ = 0.22–0.58), which is outperformed by hourly and daily averaged data ($R^2$ = 0.66–0.77), as demonstrated by Zaidan et al. (2020). The methods using the number concentration at the remaining size bins, both with or without meteorological data, show expected perfect performance ($R^2$ > 0.97). We also compared the FFNN methods with other commonly used methods and the results highlight the high accuracy and reliability of methods by means of neural networks.

Relatively poor performance of the proposed FFNN methods is found in three regions. At the lower edge (0.01< $D_p$< 0.02 μm) and the upper edge (6< $D_p$< 10 μm), the number of neighbouring size bins is limited and also the detection efficiency by the corresponding instruments is lower compared to the other size bins. Another noticeable region (0.15< $D_p$< 0.5 μm) is the overlapping section measured by the two particle sizers and the reason is because of the deficiency of merging algorithm. For all the above approaches, the poorer performance for smaller particles in the nucleation mode could be due to the fact that it is more effectively removed from the atmosphere compared to other modes (Al-Dabbous et al., 2017). An observable overestimation is also found in early morning for ultrafine particles followed by a distinct underestimation before midday. A larger derivation between the measured and the estimated number concentration is found in the winter, which might be caused by sensor drift and interference artefacts (e.g. Lewis et al., 2016; Popoola et al., 2016). Despite the high number of input predictors, the good estimation performance provides an alternative method to fill up the negative values in size distribution raw dataset, which often exist due to misconfiguration problems. Instead of removing the factually impossible data point, this way of replacing negative numbers can maintain a symmetric distribution of errors (Viskari et al., 2012) and minimise the uncertainties caused.

**Code/Data availability**

The code and data is available upon request.

**Author contribution**

TH and MZ designed the experiments and TH carried them out. PLF and OS developed the code of the proposed FFNN methods. PLF prepared the manuscript with contributions from all co-authors.

## Competing interests

The authors declare that they have no conflict of interest.

## Financial support

The work is supported by MegaSense program, the City of Helsinki Innovation Fund, Business Finland, the European Union through the Urban Innovative Action Healthy Outdoor Premises for Everyone (HOPE, project No. UIA03-240). This research is also funded by the Scientific Research Support Fund (SRF, Project Number BAS-1-2-2015) at the Jordanian Ministry of Higher Education and the Deanship of Academic Research (DAR, Project Number 1516) at the University of Jordan. This research is part of a close collaboration between the University of Jordan and the Institute for Atmospheric and Earth System Research (INAR/Physics, University of Helsinki) via ERC advanced Grant No. 742206, the European Union's Horizon 2020 research and innovation program under Grant Agreement No. 654109, the Academy of Finland Flagship funding (Project No. 337549), ERA-PLANET (www.era-planet.eu), trans-national project SMURBS (www.smurbs.eu, Grant Agreement No. 689443) funded under the EU Horizon 2020 Framework Programme, and Academy of Finland via the Center of Excellence in Atmospheric sciences and NanoBioMass (Project Number 1307537).

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

**Model optimisation**

Approach 1:
met

Approach 2:
PSD

Approach 3:
PSD + met

Model training with month separation

Adjust neural network setting
No. of layers:
1, 2
No. of neurons:
2, 4, 6, 8, 10, 15, 20, 25

Compare and decide which to use

Model evaluation

Estimation of specific bin size

Figure 1. The block diagram describing the methodology of the proposed FFNN method.

**Input layer**

**Hidden layer(s)**

**Output layer**

$x_1$

$x_2$

$x_n$

$w^{(1)}_{11}$

$w^{(1)}_{12}$

$w^{(1)}_{21}$  $w^{(1)}_{22}$

$w^{(1)}_{n1}$

$w^{(1)}_{1n}$

$w^{(1)}_{2n}$

$w^{(1)}_{n2}$

$w^{(1)}_{nn}$

$b^{(1)}_1$

$b^{(1)}_2$

$b^{(1)}_n$

$\sigma$

$\sigma$

$\sigma$

$w^{(L)}_i$

$b^{(L)}_i$

$\hat{y}$

Figure 2. Schematic diagram of a neural network with one hidden layer of sigmoid activation function.

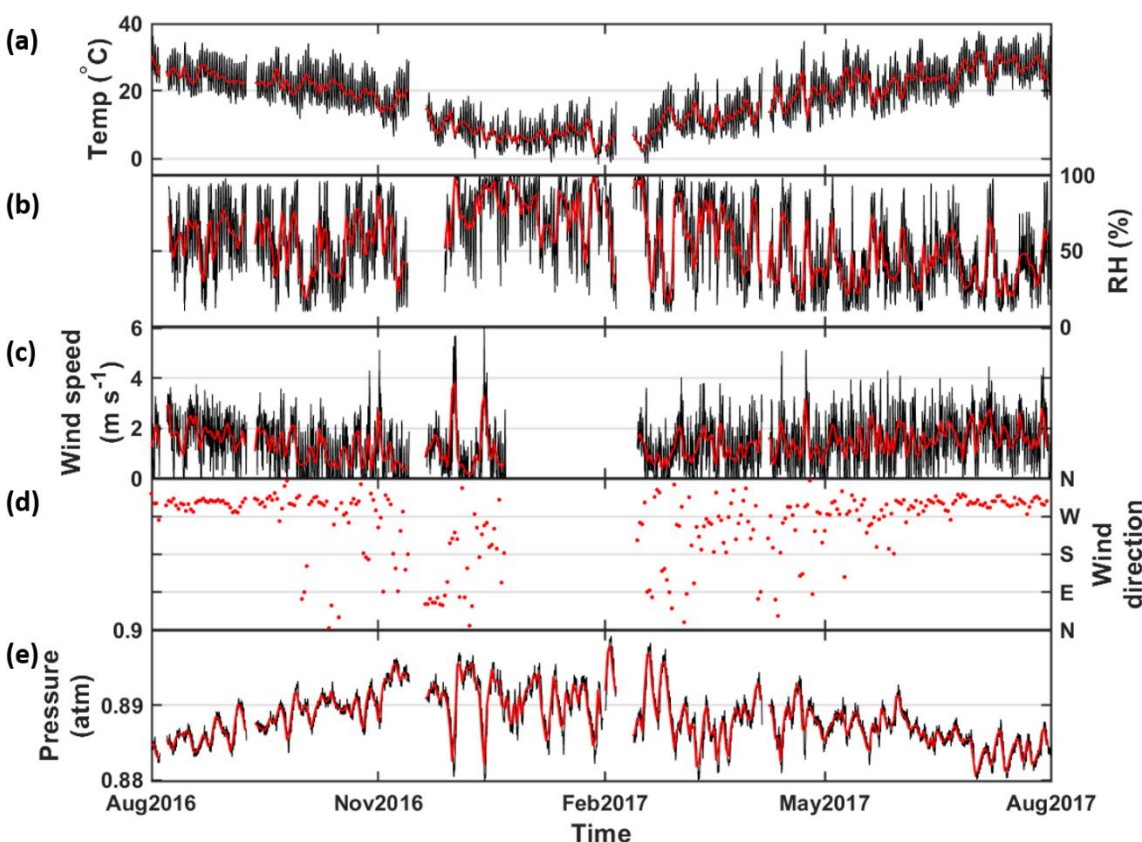

Figure 3. Timeseries of meteorological conditions during the measurement period Aug 2016–Jul 2017. (a–e) denotes temperature, relative humidity, wind speed, wind direction and air pressure, respectively. Black and red represent hourly and daily averaged data, respectively.

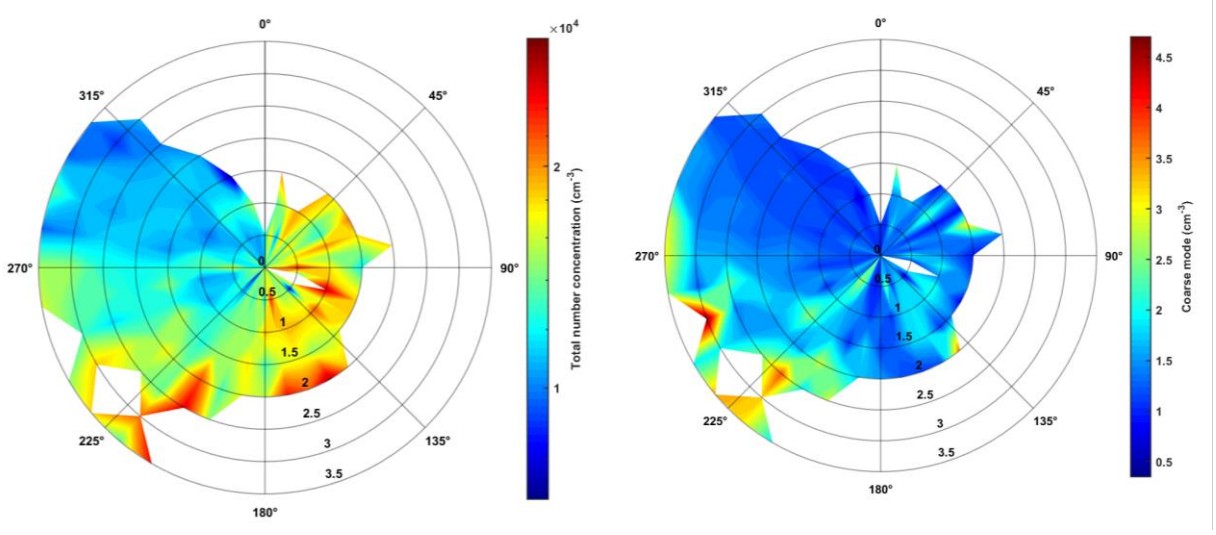

Figure 4. Windrose diagram of total particle number concentration at different direction (in theta axis) and different wind speed (in radical axis). Wind direction and wind speed data are grouped in every $10°$ and $0.5$ m s$^{-1}$. Warmer color represent higher total particle number concentration. (a) total number concentration, log scale; (b) coarse mode, linear scale. Note the color scales are different.

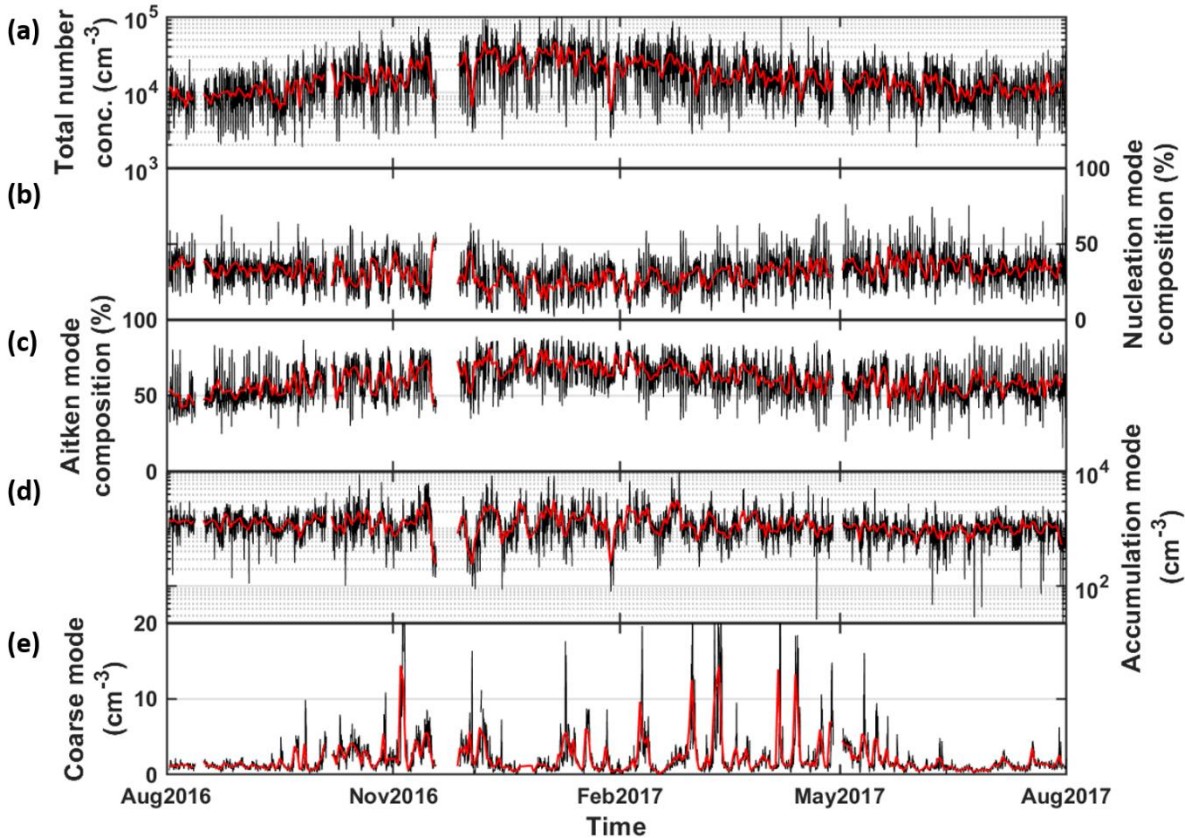

Figure 5. Timeseries of total particle number concentration (in cm$^{-3}$) of 0.01–10μm in (a). (b–c) indicate the contribution in percentage of nucleation mode and Aitken mode, respectively. (d–e) show the number concentration in accumulation mode and coarse mode, respectively. Black and red represent hourly and daily averaged data, respectively.

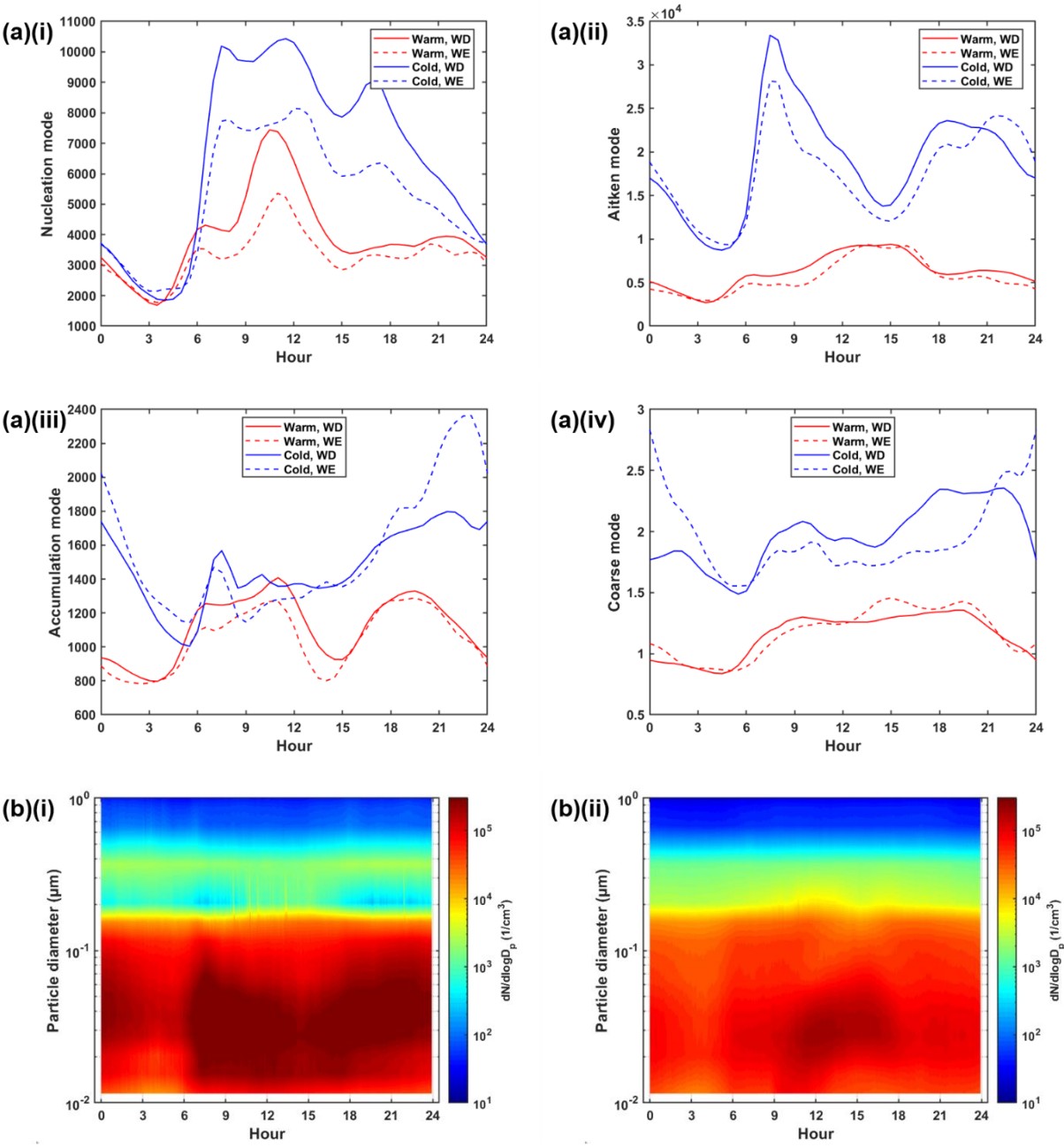

Figure 6. (a) Diurnal cycle of the (i) nucleation mode, (ii) Aitken mode, (iii) accumulation mode and (iv) coarse mode in warm (red) and cold months (blue) during workdays (solid) and weekends (dashed). (b) Particle size distribution in (i) cold and (ii) warm months, coloured by particle number concentration (cm$^{-3}$). Cold and warm months refer to December–February and June–August, respectively.

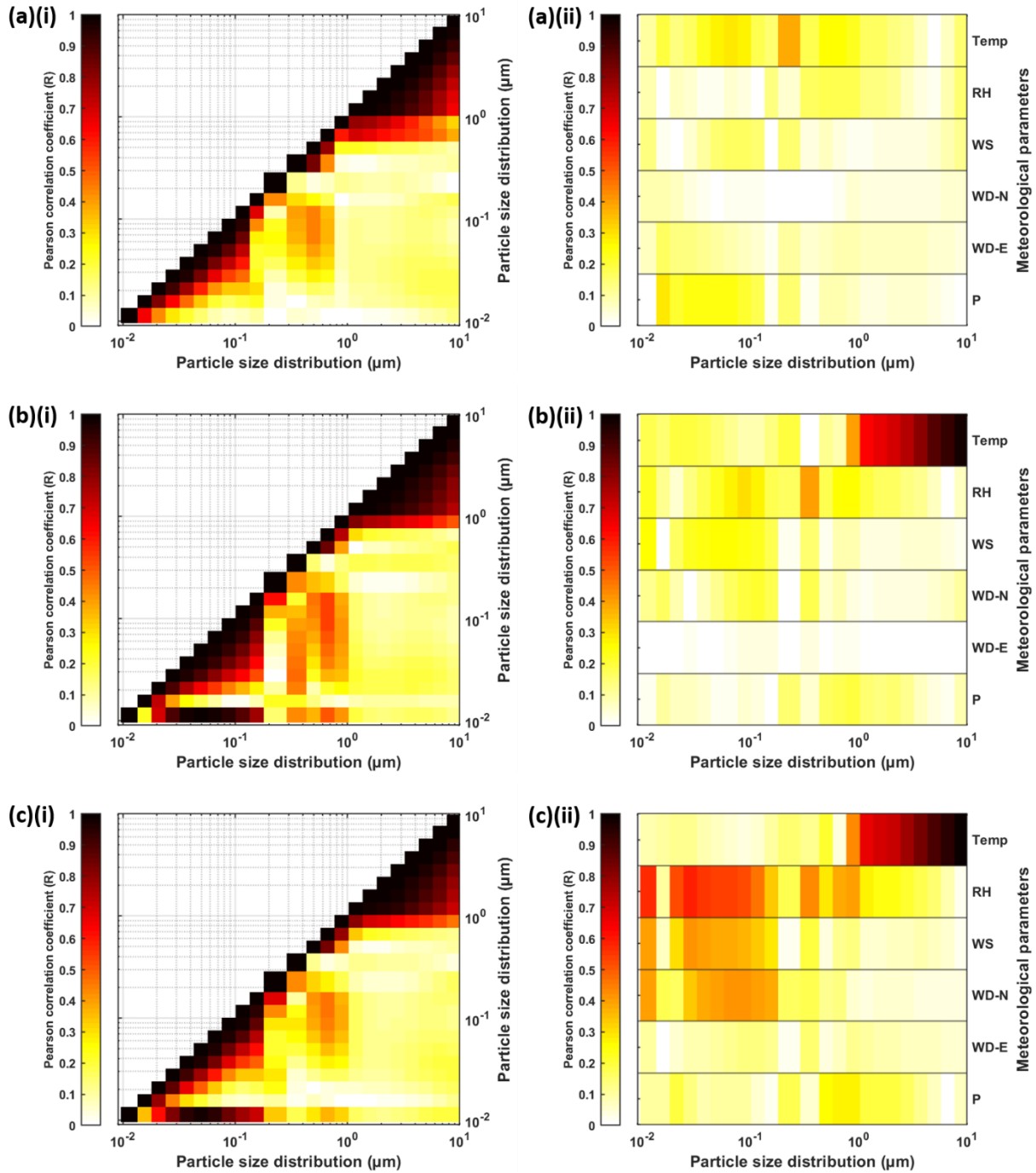

Figure 7. Matrix plots showing the Pearson correlation coefficient (R) of particle size distribution of (a) 5-min, (b) hourly, (c) daily averaging with (i) particle size distribution itself and (ii) meteorological parameters. Darker colour represents a higher correlation.

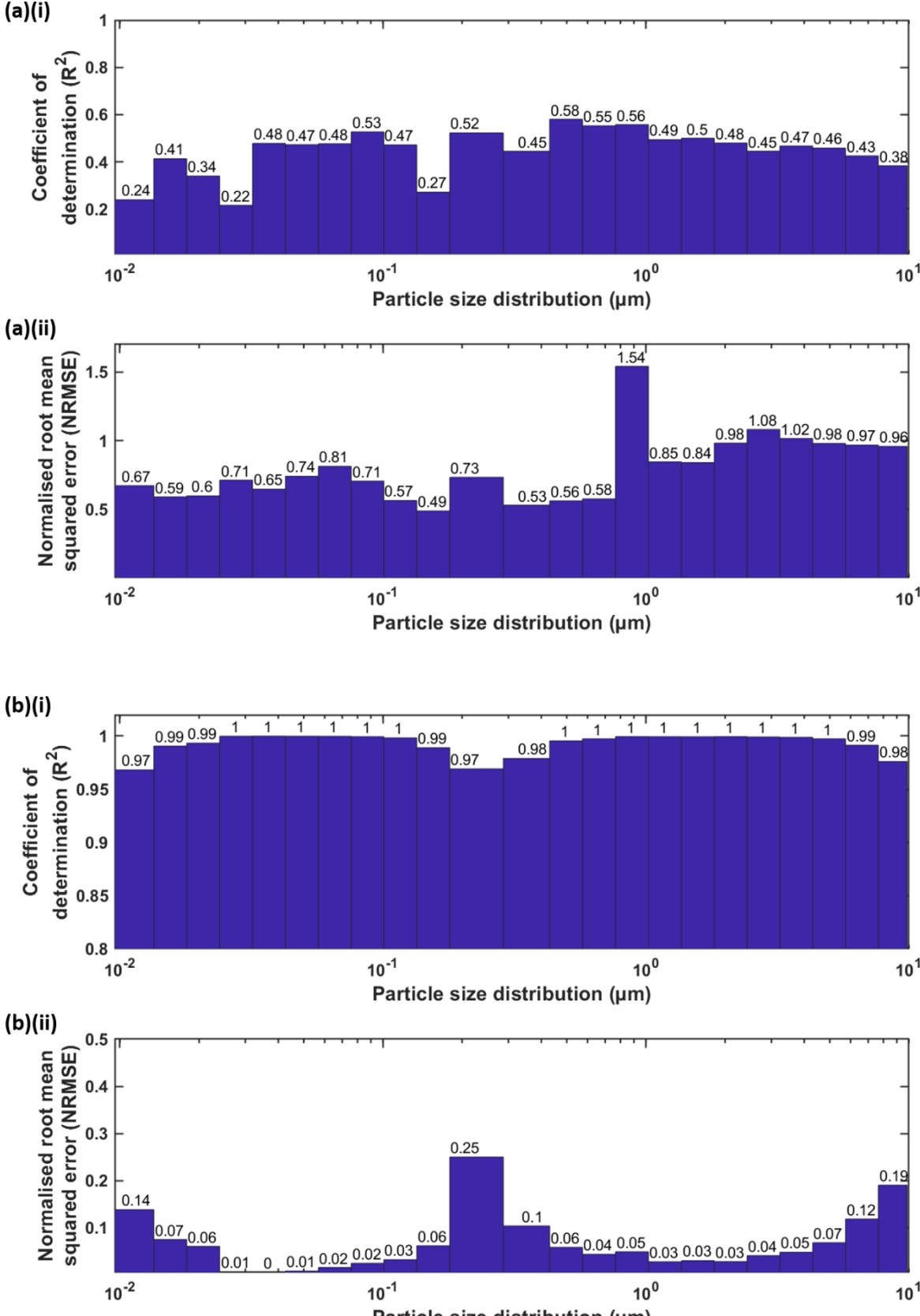

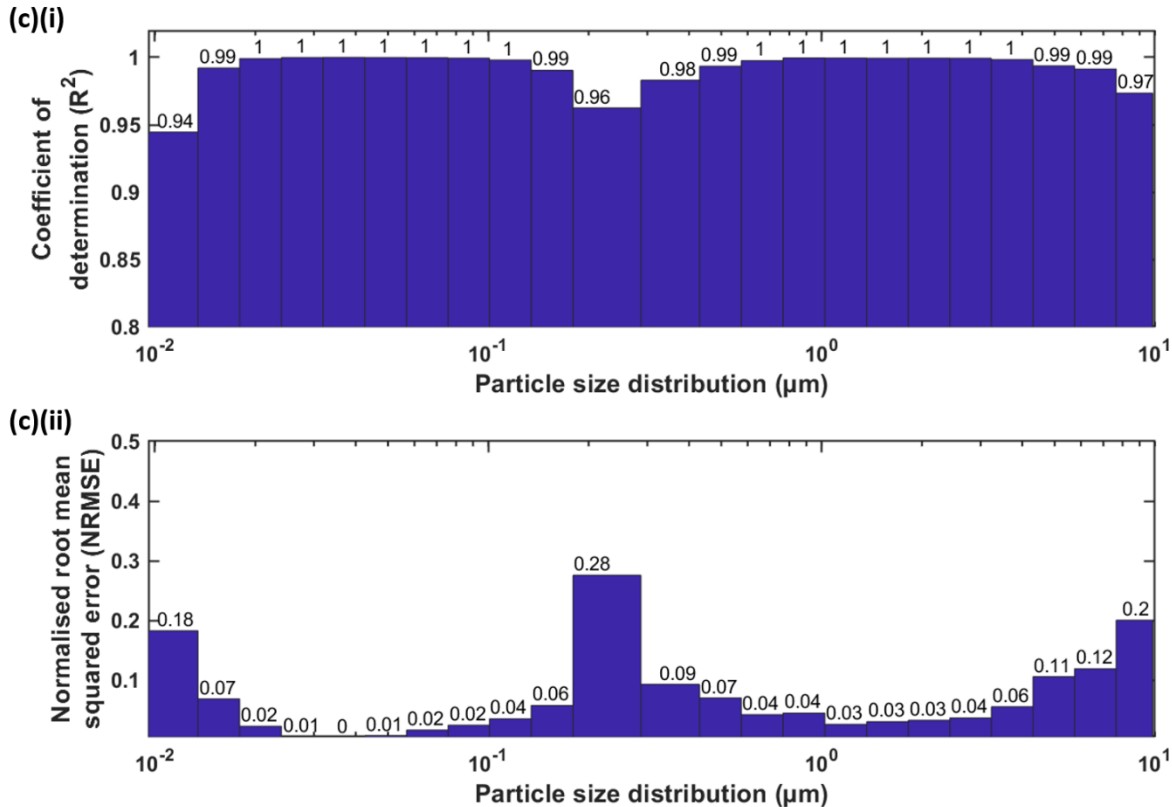

Figure 8. Bar chart showing the evaluation of FFNN approach with (a) only meteorological parameters (Approach 1, FFNN–met), (b) particle size distribution itself (Approach 2, FFNN–PSD), (c) both particle size distribution and meteorological parameters (Approach 3) as inputs. The evaluation metrics for the proposed method include (i) coefficient of determination ($R^2$) and (ii) normalised root mean squared error (NRMSE).

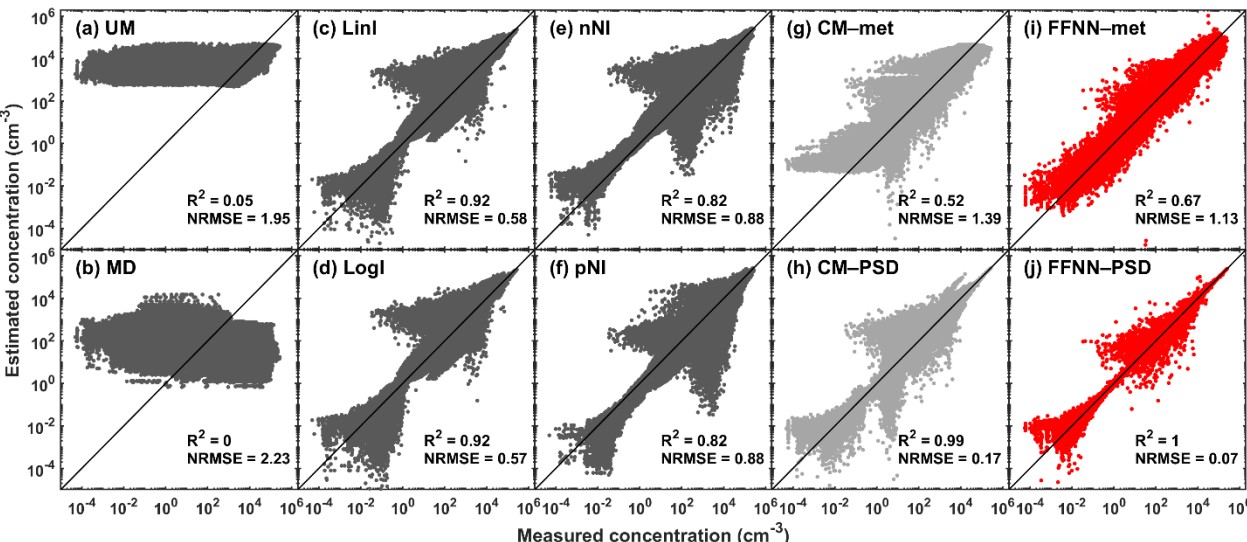

Figure 9. Scatter plots showing the estimated particle concentration (y-axis, in cm⁻³) against the measured in situ particle concentration (x-axis, in cm⁻³). (a–f) demonstrate cases of univariate methods including unconditional mean (UM), median (MD), linear interpolation (LinI), logarithmic interpolation (LogI), next neighbour interpolation (nNI) and previous neighbour interpolation (pNI), respectively, in dark grey dots. (g–h) represent multivariate methods conditional mean by regression of meteorological parameters and other particle size number concentrations as inputs (CM–met and CM–PSD, respectively) in light grey dots. (i–j) showcase the proposed feed-forward neural network with meteorological parameters and other particle size number concentrations as inputs (FFNN–met and FFNN–PSD, respectively) in red dots. The black solid line is 1:1 line which gives a reference of perfect estimation. The coefficient of determination ($R^2$) and the normalised root-mean-square error (NRMSE) of each method for all particle size bins are printed on the corresponding subplots.

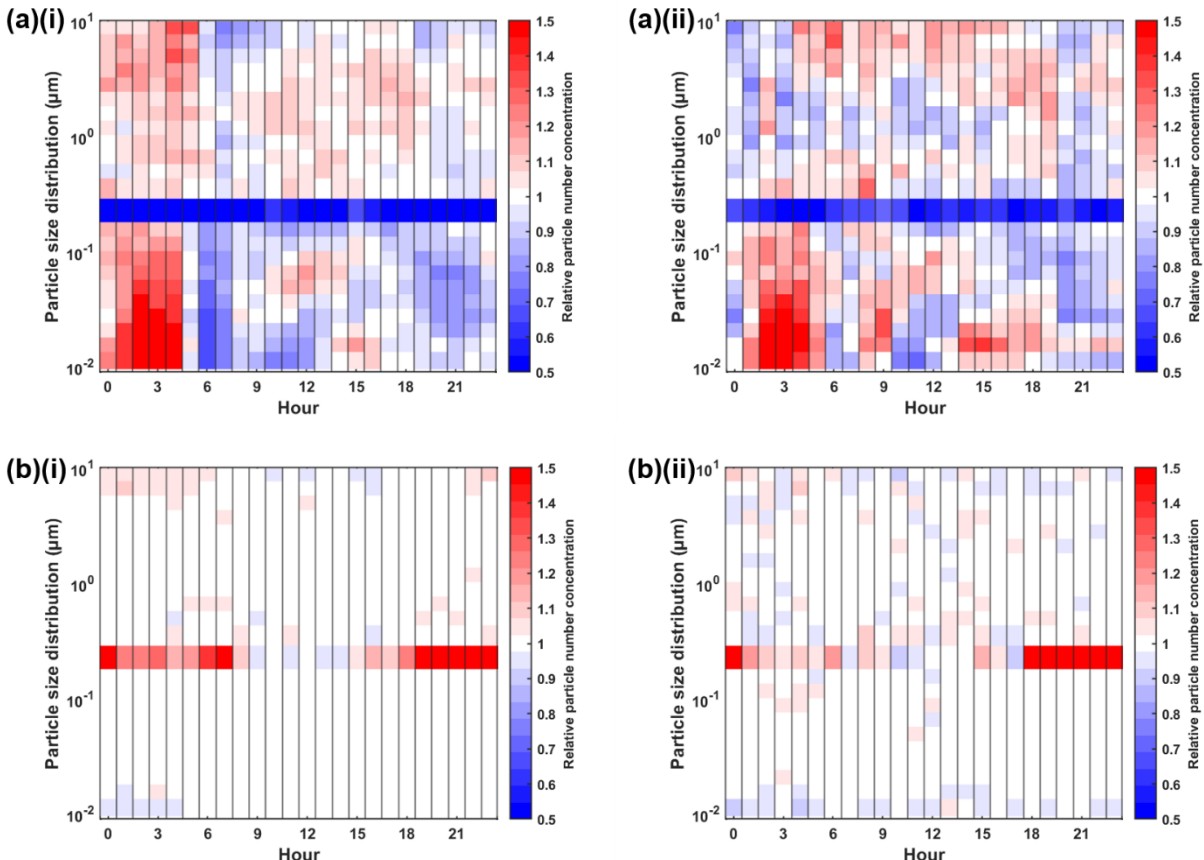

Figure 10. Heatmap showing the hourly median relative particle number concentration of the approach with (a) meteorological parameters (Approach 1, FFNN–met) and (b) particle size distribution (Approach 2, FFNN–PSD) as inputs across different hours of a day (i) in workdays and (ii) in weekends. The relative particle number concentration is defined as estimated concentration with respect to measured concentration. Red colour show overestimation while blue show underestimation.

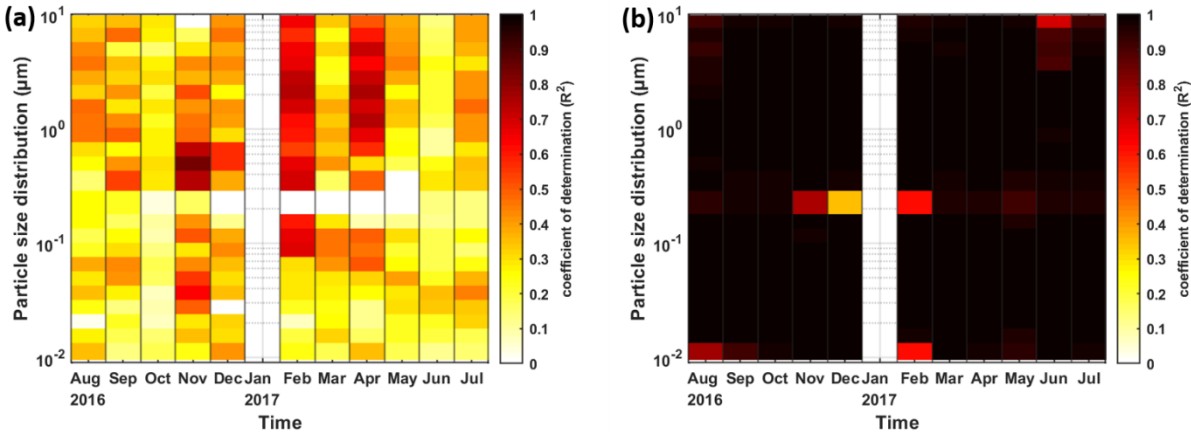

Figure 11. Heatmap showing the coefficient of determination ($R^2$) of the approach with (a) meteorological parameters (Approach 1, FFNN–met) and (b) particle size distribution (Approach 2, FFNN–PSD) as inputs for different months at different size bins. Darker colour represents a higher $R^2$.

Table 1. Table showing the descriptive statistics (in cm$^{-3}$) of total number concentration, nucleation mode, Aitken mode,
accumulation mode and coarse mode. The statistical values include mean, standard deviation, and percentile (10%, 25%,
50%, 75% and 90%).

| | Mean | std | 10% | 25% | 50% | 75% | 90% |
|---|---|---|---|---|---|---|---|
| Total ($\times 10^4$) | 1.70 | 1.26 | 0.57 | 0.85 | 1.35 | 2.16 | 3.31 |
| Nucleation ($\times 10^4$) | 0.48 | 0.32 | 0.16 | 0.26 | 0.41 | 0.63 | 0.90 |
| Aitken ($\times 10^4$) | 1.09 | 1.01 | 0.29 | 0.45 | 0.77 | 1.37 | 2.35 |
| Accumulation ($\times 10^4$) | 0.13 | 0.08 | 0.05 | 0.08 | 0.11 | 0.15 | 0.21 |
| Coarse | 2.13 | 2.80 | 0.55 | 0.84 | 1.29 | 2.33 | 4.3 |

Table 2. Table showing the best configuration in the form of (the number of layers; the number of neurons) for the
approach by meteorological parameters (FFNN–met) and the number concentration at the other size bins (FFNN–PSD)
as inputs. Mean absolution error (MAE, in cm$^{-3}$), coefficient of determination ($R^2$) and normalised root-mean-square error
(NRMSE) are listed for different size bins on each row. The last row concludes the overall selection of the approach with
the best configuration and its corresponding evaluation metrics.

| Particle size (µm) | Approach 1 (FFNN–met) | | | | Approach 2 (FFNN–PSD) | | | |
|---|---|---|---|---|---|---|---|---|
| | Best setting | MAE (cm$^{-3}$) | $R^2$ | NRMSE | Best setting | MAE (cm$^{-3}$) | $R^2$ | NRMSE |
| 0.012 | 2; 10 | 2640 | 0.20 | 0.69 | 2; 10 | 334 | 0.99 | 0.11 |
| 0.015 | 2; 15 | 4850 | 0.42 | 0.59 | 2; 8 | 216 | 1.00 | 0.031 |
| 0.021 | 2; 15 | 6120 | 0.38 | 0.58 | 2; 15 | 97.8 | 1.00 | 0.014 |
| 0.027 | 2; 15 | 8470 | 0.41 | 0.62 | 1; 25 | 34.0 | 1.00 | 0.0032 |
| 0.037 | 2; 20 | 8240 | 0.46 | 0.66 | 2; 15 | 26.3 | 1.00 | 0.0024 |
| 0.049 | 2; 15 | 6610 | 0.48 | 0.74 | 2; 25 | 33.7 | 1.00 | 0.0049 |
| 0.066 | 2; 15 | 4690 | 0.46 | 0.83 | 2; 10 | 56.7 | 1.00 | 0.013 |
| 0.088 | 2; 15 | 3040 | 0.52 | 0.71 | 2; 4 | 66.2 | 1.00 | 0.018 |
| 0.12 | 2; 15 | 1810 | 0.52 | 0.54 | 2; 8 | 63.1 | 1.00 | 0.021 |
| 0.15 | 2; 10 | 917 | 0.29 | 0.49 | 2; 15 | 72.5 | 0.99 | 0.052 |
| 0.21 | 2; 6 | 327 | 0.55 | 0.71 | 2; 8 | 114 | 0.91 | 0.31 |
| 0.37 | 2; 10 | 95.8 | 0.43 | 0.54 | 2; 20 | 12.9 | 0.99 | 0.072 |
| 0.49 | 2; 15 | 12.1 | 0.50 | 0.61 | 2; 25 | 0.9630 | 1.00 | 0.043 |
| 0.66 | 2; 15 | 3.03 | 0.58 | 0.56 | 2; 15 | 0.1995 | 1.00 | 0.029 |
| 0.88 | 2; 15 | 5.65 | 0.62 | 1.43 | 2; 10 | 0.2202 | 1.00 | 0.040 |
| 1.17 | 2; 15 | 1.43 | 0.53 | 0.81 | 2; 8 | 0.0680 | 1.00 | 0.026 |
| 1.56 | 2; 20 | 1.44 | 0.54 | 0.81 | 2; 8 | 0.0816 | 1.00 | 0.031 |
| 2.08 | 2; 15 | 1.84 | 0.49 | 0.97 | 2; 8 | 0.0825 | 1.00 | 0.028 |
| 2.77 | 2; 15 | 1.02 | 0.44 | 1.09 | 1; 4 | 0.0573 | 1.00 | 0.037 |
| 3.70 | 2; 15 | 0.52 | 0.41 | 1.07 | 1; 8 | 0.0329 | 1.00 | 0.046 |
| 4.92 | 2; 15 | 0.28 | 0.44 | 1.00 | 1; 4 | 0.0254 | 1.00 | 0.068 |
| 6.56 | 2; 9 | 0.11 | 0.42 | 0.97 | 1; 6 | 0.0206 | 0.99 | 0.13 |
| 8.75 | 2; 10 | 0.060 | 0.39 | 0.95 | 2; 6 | 0.0169 | 0.98 | 0.20 |
| overall | 2; 15 | 2120 | 0.67 | 1.13 | 2; 10 | 76.6 | 0.999 | 0.067 |

662

Table 3. Table showing the comparison of different estimation methods, including unconditional mean (UM, column 2), median (MD, column 3), linear interpolation (LinI, column 4), logarithmic interpolation (LogI, column 5), next neighbour interpolation (nNI, column 6), previous neighbour interpolation (pNI, column 7), conditional mean by regression of meteorological parameters and other particle size number concentrations as inputs (CM–met and CM–PSD, column 8 and 9, respectively) and the feed-forward neural network with meteorological parameters and other particle size number concentrations as inputs (FFNN–met and FFNN–PSD, column 10 and 11, respectively). The coefficient of determination ($R^2$) of each method are listed for different size bins on each row. Negative $R^2$ are represented as '0' to indicate poor accuracy at the particular particle size bin while 'NA' is used to represent the data is not available. The last row concludes the overall evaluation metrics.

| Particle size (µm) | Methods/ $R^2$ | | | | | | | | | |
|---|---|---|---|---|---|---|---|---|---|---|
| | UM | MD | LinI | LogI | nNI | pNI | CM –met | CM –PSD | FFNN –met | FFNN –PSD |
| 0.012 | 0 | 0 | 0 | 0 | 1.00 | NA | 0.04 | 0.91 | 0.20 | 0.99 |
| 0.015 | 0 | 0 | 0.66 | 0.71 | 0 | 0.49 | 0.14 | 0.85 | 0.42 | 1.00 |
| 0.021 | 0 | 0 | 0.92 | 0.91 | 0.62 | 0.33 | 0.1 | 1.00 | 0.38 | 1.00 |
| 0.027 | 0 | 0 | 0.91 | 0.93 | 0.69 | 0.90 | 0.11 | 1.00 | 0.41 | 1.00 |
| 0.037 | 0 | 0 | 0.97 | 0.97 | 0.91 | 0.85 | 0.12 | 1.00 | 0.46 | 1.00 |
| 0.049 | 0 | 0 | 0.98 | 0.99 | 0.80 | 0.80 | 0.13 | 1.00 | 0.48 | 1.00 |
| 0.066 | 0.14 | 0 | 0.96 | 0.97 | 0.66 | 0.81 | 0.14 | 1.00 | 0.46 | 1.00 |
| 0.088 | 0.31 | 0 | 0.97 | 0.98 | 0.60 | 0.64 | 0.12 | 1.00 | 0.52 | 1.00 |
| 0.12 | 0.41 | 0 | 0.92 | 0.96 | 0 | 0 | 0.07 | 1.00 | 0.52 | 1.00 |
| 0.15 | 0 | 0 | 0 | 0.20 | 0 | 0 | 0.03 | 0.97 | 0.29 | 0.99 |
| 0.21 | 0 | 0 | 0 | 0 | 0 | 0 | 0.24 | 0.65 | 0.55 | 0.91 |
| 0.37 | 0 | 0 | 0 | 0 | 0 | 0 | 0.04 | 0.9 | 0.43 | 0.99 |
| 0.49 | 0 | 0 | 0 | 0 | 0 | 0 | 0.06 | 0.97 | 0.50 | 1.00 |
| 0.66 | 0 | 0 | 0 | 0 | 0 | 0 | 0.07 | 0.96 | 0.58 | 1.00 |
| 0.88 | 0 | 0 | 0.20 | 0.19 | 0.23 | 0.11 | 0.09 | 0.76 | 0.62 | 1.00 |
| 1.17 | 0 | 0 | 0 | 0 | 0 | 0.99 | 0.04 | 1.00 | 0.53 | 1.00 |
| 1.56 | 0 | 0 | 0.97 | 0.97 | 0.99 | 0.85 | 0.04 | 1.00 | 0.54 | 1.00 |
| 2.08 | 0 | 0 | 0.84 | 0.83 | 0.91 | 0.67 | 0.03 | 1.00 | 0.49 | 1.00 |
| 2.77 | 0 | 0 | 0.90 | 0.96 | 0 | 0.60 | 0.02 | 1.00 | 0.44 | 1.00 |
| 3.70 | 0 | 0 | 0.76 | 0.87 | 0 | 0.62 | 0.02 | 1.00 | 0.41 | 1.00 |
| 4.92 | 0 | 0 | 0.85 | 0.94 | 0 | 0.41 | 0.02 | 1.00 | 0.44 | 1.00 |
| 6.56 | 0 | 0 | 0.27 | 0.55 | 0 | 0.57 | 0.03 | 0.99 | 0.42 | 0.99 |
| 8.75 | 0 | 0 | 0 | 0 | NA | 1.00 | 0.05 | 0.97 | 0.39 | 0.98 |
| overall | 0.05 | 0 | 0.92 | 0.92 | 0.82 | 0.82 | 0.52 | 0.99 | 0.67 | 1.00 |

Table 4. Table showing the comparison of different estimation methods, including unconditional mean (UM, column 2), median (MD, column 3), linear interpolation (LinI, column 4), logarithmic interpolation (LogI, column 5), next neighbour interpolation (nNI, column 6), previous neighbour interpolation (pNI, column 7), conditional mean by regression of meteorological parameters and other particle size number concentrations as inputs (CM–met and CM–PSD, column 8 and 9, respectively) and the feed-forward neural network with meteorological parameters and other particle size number concentrations as inputs (FFNN–met and FFNN–PSD, column 10 and 11, respectively). The normalised root-mean-square error (NRMSE) of each method are listed for different size bins on each row. The last row concludes the overall evaluation metrics.

| Particle size (µm) | Methods/ NRMSE | | | | | | | | | |
|---|---|---|---|---|---|---|---|---|---|---|
| | UM | MD | LinI | LogI | nNI | pNI | CM –met | CM –PSD | FFNN –met | FFNN –PSD |
| 0.012 | 0.84 | 1.24 | 1.62 | 1.73 | NA | 1.62 | 0.74 | 0.23 | 0.69 | 0.11 |
| 0.015 | 0.92 | 1.26 | 0.45 | 0.42 | 0.79 | 0.55 | 0.72 | 0.30 | 0.59 | 0.03 |
| 0.021 | 0.91 | 1.24 | 0.21 | 0.22 | 0.46 | 0.61 | 0.70 | 0.02 | 0.58 | 0.01 |
| 0.027 | 1.04 | 1.28 | 0.24 | 0.22 | 0.46 | 0.25 | 0.77 | 0 | 0.62 | 0 |
| 0.037 | 1.08 | 1.34 | 0.15 | 0.15 | 0.27 | 0.35 | 0.85 | 0 | 0.66 | 0 |
| 0.049 | 1.09 | 1.43 | 0.13 | 0.12 | 0.46 | 0.46 | 0.95 | 0 | 0.74 | 0 |
| 0.066 | 1.04 | 1.50 | 0.23 | 0.18 | 0.66 | 0.49 | 1.04 | 0.01 | 0.83 | 0.01 |
| 0.088 | 0.84 | 1.42 | 0.16 | 0.13 | 0.65 | 0.61 | 0.96 | 0.02 | 0.71 | 0.02 |
| 0.12 | 0.59 | 1.25 | 0.22 | 0.16 | 0.86 | 0.80 | 0.74 | 0.03 | 0.54 | 0.02 |
| 0.15 | 1.59 | 1.13 | 0.66 | 0.53 | 1.64 | 0.96 | 0.58 | 0.10 | 0.49 | 0.05 |
| 0.21 | 11.6 | 1.61 | 3.7 | 3.24 | 4.93 | 1.53 | 1.26 | 0.85 | 0.71 | 0.31 |
| 0.37 | 23.8 | 1.42 | 1.35 | 1.12 | 3.12 | 1.06 | 0.70 | 0.22 | 0.54 | 0.07 |
| 0.49 | 185 | 14.4 | 4.16 | 3.53 | 7.98 | 1.00 | 0.83 | 0.15 | 0.61 | 0.04 |
| 0.66 | 672 | 54.5 | 2.42 | 2.32 | 3.62 | 2.79 | 0.82 | 0.17 | 0.56 | 0.03 |
| 0.88 | 485 | 39.4 | 2.06 | 2.07 | 2.02 | 2.18 | 2.20 | 1.12 | 1.43 | 0.04 |
| 1.17 | 1750 | 143 | 4.45 | 3.88 | 7.84 | 0.11 | 1.16 | 0.07 | 0.81 | 0.03 |
| 1.56 | 1750 | 143 | 0.19 | 0.22 | 0.11 | 0.46 | 1.16 | 0.05 | 0.81 | 0.03 |
| 2.08 | 1510 | 124 | 0.54 | 0.57 | 0.40 | 0.78 | 1.34 | 0.04 | 0.97 | 0.03 |
| 2.77 | 2880 | 236 | 0.47 | 0.30 | 1.48 | 0.92 | 1.43 | 0.04 | 1.09 | 0.04 |
| 3.70 | 5750 | 472 | 0.69 | 0.50 | 1.83 | 0.86 | 1.38 | 0.05 | 1.07 | 0.05 |
| 4.92 | 11000 | 902 | 0.51 | 0.34 | 1.64 | 1.02 | 1.32 | 0.09 | 1.00 | 0.07 |
| 6.56 | 27100 | 2220 | 1.09 | 0.86 | 2.51 | 0.83 | 1.26 | 0.12 | 0.97 | 0.13 |
| 8.75 | 52600 | 4320 | 4.95 | 3.33 | 1.62 | NA | 1.2 | 0.21 | 0.95 | 0.20 |
| overall | 1.95 | 2.23 | 0.58 | 0.57 | 0.88 | 0.88 | 1.39 | 0.17 | 1.13 | 0.07 |