# Peer review of "Data imputation in in situ measured particle size distributions by"

_Atmospheric Measurement Techniques, 2021_

## Author Comment (AC1)

We show our gratitude to Anonymous Referee #2 for his constructive comments. We have revised the manuscript accordingly. Please find our point-to-point responses below.

**Response to Anonymous Referee #2's comments**

To my understanding, this manuscript proposes a neural network method to fill for missing/invalid values in particle size distributions (PSDs) measured in situ, based on correct values measured at other size bins and on knowledge of meteorological parameters. They train a number of neural networks in which one out of 23 bins in the considered size distributions is considered missing, and is predicted based on meteorological parameters and/or on the other 22 size bins.

The study seems to show that, if PSD values are known at 22 size bins, meteorological data do not add much information for the prediction of the missing value. This finding does not sound particularly surprising to me, as I would expect PSDs to be relatively smooth, and knowing their values at 22 size bins should indeed provide a good deal of information about its value at some neighbouring bins.

MAIN COMMENTS

• While the technical work of setting up and training the neural networks looks correct as far as I can see, the main problems I see with this paper are the following:

1)    The first problem is that I had to carefully peruse the manuscript several times before understanding what the main goal exactly was. This means that the Authors should be more upfront in describing the goals of the study and its setup in the introduction. In particular, looking at the Introduction from Line 146 onwards, I had the impression that the scope was much broader than just interpolating a PSD at some missing values. Maybe also the title can be made more informative: instead of reading "Neural network modelling … on other particle sections, etc." it may be something like "Replacement of missing values in in situ measured particle size distributions by means of neural networks". This would give the reader a more immediate idea of what they are going to find in this paper.

Response: We understand that it might be a bit misleading to put 'neural network modelling' as the title. Considering the suggestion by the referee, we change the title as 'Data imputation in in situ measured particle size distributions by means of neural networks'. Although the concept of the paper is the replacement of negative data from raw values of particle sizer instruments, the operational applicability is wider than that because the proposed neural network method can serve as a supplement to better the current built-in algorithm of in situ particle sizer instruments. To emphasise this significant goal, we cut the unnecessary parts in the introduction and reiterate this ultimate goal throughout the manuscript.

2)    The second problem is that I am not entirely sure what the scientific significance of this study is. Training a neural network to fill in a single value in a PSD based on knowledge of other 22 values looks like a rather standard technical exercise to me. I am not sure if it makes sense to write a scientific paper about this, but since I am not part of the in situ measurement community I may not be the most appropriate person to comment on this. In general, when you use a NN to fill for missing values in the PSD you are basically developing a functional relationship between the PSD value at the missing bin and its values at other bins, and it is quite obvious that this relationship will work well or less well depending on how the real PSD actually behaves. I am not sure what general conclusions can be drawn from the analysis presented in section 4, except for the one – widely expected in my opinion – that knowing already the PSD at almost all size bins helps a lot in predicting the missing value, compared to only knowing some meteorological information.

Response: In the field of in situ aerosol measurement, we have identified the problem that the built-in algorithm in instruments could be sometimes ill-configured and generate meaningless numbers, such as negative values especially when the actual number concentration is low. Currently, while some experts use the built-in inversion algorithm, some develop their own inversion methods. However, the tailor-made inversion algorithms have their drawbacks. Therefore, the use of built-in inversion in aid of neural networks could be a good alternative. Considering the other suggestion the referee gave, we compare this FFNN method with other simpler methods and it apparently demonstrates a better performance in terms of accuracy and reliability even though all those simpler methods also only use PSD for interpolation.

• To have a clearer idea of the added value of using a NN to replace a missing size bin in a PSD, it would be interesting to see how the NN method performs compared to a simpler approaches such as linear interpolation.

Response: Thank you for the suggestion. We agree that comparing the feed-forward neural network (FFNN) methods with other simpler approaches would be a good added value to this paper. For this, we compute the unconditional mean (UM), linear interpolation (LinI), logarithmic interpolation (LogI), next neighbour interpolation (nNI), previous neighbour interpolation (pNI) and conditional mean by regression of meteorological parameters and other particle size number concentrations as inputs (CM–met and CM–PSD, respectively). The comparison of $R^2$ and NRMSE is presented as Table 3 and Table 4. We also summarise the results as texts to the manuscript in Section 4.1. FFNN models outperform all the univariate methods and FFNN–PSD also shows better accuracy than CM–PSD.

• While it is fair to say that a NN predicting PSD solely based on meteorological inputs constitutes indeed a predictive model (you can try to predict the PSD anytime and anywhere you have the needed meteorological input), I am not sure the same holds for the NN that also uses measurements of the PSD itself. I mean, if you are not measuring the PSD I guess you cannot operate such NN. Therefore, I would say that the NN using PSD values as input is not a "model to estimate the PSD" but just a "NN to interpolate missing values in the PSD". Can you please elaborate on the impact that using PSD values as input has on the operational applicability of your NN?

Response: This is very important that you have pointed out the operational applicability of the proposed neural network models (FFNN). We understand that this proposed method uses the measurements of the PSD itself and it is apparently an imputation method instead of a predictive model. However, we believe that, due to the shortcomings of the built-in algorithm in particle sizer instruments, this FFNN method can improve the algorithm and generate meaningful numbers when measuring the particle size distribution. For this, we further elaborate the significant impacts on the operational applicability of the FFNN methods that use PSD as inputs. Besides, to avoid confusion, we replace the word 'models' with 'methods' throughout the manuscript.

OTHER COMMENTS

• P1, L18, "which are able to deposit…". What is the subject of this sentence?

Response: The sentence is now revised as '… fractionated size distribution, in which particles of different diameters are able to deposit …'

• P1, L21, "invert" -> "solve"

Response: Revised accordingly.

• P2, L47. I guess Dp is the particle diameter. You should state this explicitly in the text.

Response: 'Dp' is the particle diameter in this manuscript. This explanations are now stated several times in the current version. And it is now revised as '$D_p$' as suggested by another referee.

• P3, L109 and L118. What are these "kernel functions", and what does it mean that they are "not optimally configured"?

Response: The kernel functions describe the probability of particles of a certain size being measured at a certain flow rate. Thus the measured concentration in a size bin is the sum of the actual concentrations multiplied by the kernel functions. For example in Gaussian inversion method, the width and height of the kernel functions were chosen so that they resemble the difference between the measured activation curves, taking the detection efficiency into account (Lehtipalo et al., 2014). They can be 'not optimally configured' in case of a change in measured activation curves or a drop in detection efficiency. A short sentence is added to the introduction paragraph to explain this in detail.

• P3, L112, and P5, L184. It looks to me like the acronym CPC has not been defined anywhere.

Response: The acronym CPC represents Condensation Particle Counter and the definition is now included in the text.

• P4, L146, "objectives" -> "objective"

Response: Revised accordingly.

• P5, L182 and L185. I think the acronym DMA is also not defined anywhere in the paper.

Response: The acronym DMA represents Differential Mobility Analyzer and the definition is now included in the text.

• P6, L211. I guess you mean "linearly interpolated in time" here

Response: Revised accordingly.

• Section 2.3. Please summarize the inputs you used in your NNs (which and how many), preferably in a table. This information is hard to find in the paper as it is now, and should be readily available to the reader.

Response: Part of the information (the inputs of the three approaches of neural network methods) was shown in Figure 1. Considering the suggestion, we update the figure by mentioning the number of layers and number of neurons we test when developing the neural network methods.

• P6, L226. What do you mean by "generate a signal or remain silent"? Are you using a binary threshold activation function? Later on you say that you are using a sigmoid.

Response: We use sigmoid as the activation function. To avoid confusion, we remove 'generate a signal or remain silent'. Thank you for the bringing this up.

• P10, several instances. What do you mean by "mutual size distribution"? "mutual" between what?

Response: 'Mutual' here meant mutual relationship between the size sections in the aerosol population. To avoid confusion, we remove some of them and insert the above explanation in Section 4.1.

• P11, L384, I guess "which leads to a lower predictably" contains some typo. What do you actually mean?

Response: The correct word sentence is 'which leads to a lower predictability', which is now corrected in the paragraph.

- P12, L455, by "trivial" do you mean "negligible"?

Response: Thank you for suggesting a better word in this sentence. This is now corrected accordingly.

- P13, L466, "ill-configured" -> "misconfiguration" ?

Response: Thank you for suggesting a better word in this sentence. This is now corrected accordingly.

---

## Author Comment (AC2)

We show our gratitude to Anonymous Referee #1 for his constructive comments. We have revised the manuscript accordingly. Please find our point-to-point responses below.

**Response to Anonymous Referee #1's comments**

The aim of the study is to predict ambient particle number size distributions from 0.01 to 10 μm using meteorological data and partial size-fractionated particle data. The authors provide a good overview of the lack of particle size distribution measurements in the MENA region and application of machine learning models for atmospheric aerosol research. The authors utilized an urban aerosol dataset collected in Amman, Jordan via tandem SMPS/OPS measurements. 70% was allocated as the training set and 30% for the testing set for the feedforward neural network (FFNN) modeling analysis. Seasonal and diurnal trends in the SMPS/OPS data are concisely described. The authors provide a thorough breakdown of the performance of the three modeling approaches considered: (1) meteorological data, (2) partial particle data from remaining size bins, and (3) a combination of (1) and (2). The authors found meteorological data to be of limited value to their modeling process and noted issues with predicting particle size distributions from 0.01 to 10 μm based on partial size-fractionated particle data. The study is of value for those conducting ambient aerosol measurements with the TSI NanoScan SMPS 3910 and TSI OPS 3330, which is a common configuration with noted limitations. The authors identified size fractions where their model did not perform well, notably near the lower limit of detection for the SMPS, near the upper limit of detection for the OPS, and in the overlapping region between the two instruments. The study provides a new way for handling particle number size distribution data collected from the TSI NanoScan SMPS 3910 and TSI OPS 3330 that forgos deletion of negative values in the dataset. The proposed FFNN modeling approach can be used to predict particle number size distributions in urban environments similar to the one considered in the study.

Response: Thank you for the good summary of our manuscript.

Specific Comments

1) Pg. 5, Ln. 170: what refractive index (default or otherwise) was used for processing the data from the TSI OPS 3330?

Response: Default TSI setting was used.

2) Pg. 5, Ln. 175: please describe the configuration of the aerosol inlet assembly. Also: was a diffusion dryer used? I am curious if the authors encountered any issues with SMPS/OPS sampling at high RH (RH of 100% reported on pg. 8)?

Response: The inlet consisted of copper tubing with a diffusion drier (TSI 3062-NC). The penetration efficiency and losses in the inlet and the diffusion drier were estimated experimentally in the lab. The sentence is now inserted in Section 2.1.

3) Pg. 5, Ln. 180: it would be helpful to explain some of the limitations of the unipolar charger and radial DMA used within the TSI NanoScan SMPS 3910 as it pertains to the counting efficiency issues discussed in this section.

Response: The SMPS is designed to measure the size distribution within 0.01–0.45 μm. However, due to limitations related to the unipolar charger, the upper limit of the size distribution was 0.25 μm instead of 0.45 μm. The complete size distribution 0.01–0.1 μm was constructed by combining the distributions

measured with the SMPS (after removing the last two bins) and the OPS (after removing the first bin). The combination was based on interpolation to match the two distributions.

4) Pg. 20, Fig. 6: please specify the cold and warm months in the figure caption.

Response: Cold and warm months refer to Dec–Feb and Jun–Aug, respectively. This is now included in the figure caption.

Technical Corrections

1) General: consider writing "Dp" with "p" as a sub-script.

Response: We have now corrected all 'Dp' to '$D_p$'.

2) Pg. 1, Ln. 1: consider making "particle size distribution" plural in the title.

Response: Considering also the title suggestion by referee #2, we have now the title as 'Data imputation in in situ measured particle size distributions by means of neural networks'.

3) Pg. 1, Ln. 16: please clarify if you mean size-integrated particle mass concentrations, e.g. PM2.5, PM10.

Response: Thank you for the suggestion and we specify the particle mass concentrations as size-integrated as suggested.

---

## Author Response (AR2)

Response to Editor

Thank you for addressing the reviewer comments. I believed the manuscript improved a lot.

The addition of tables 3&4 is very important, as it shows the added value of the proposed method compared to standard methods.

Response: Thank you for the positive comments.

However, this needs more explanation. The improvement is more than I would expect given that PSDs are typically smooth. To guide the discussion, it would be informative to include 1 or more figures showing the reconstructed PSD for different methods.

Response: Thank you for the suggestion. I have now made a new figure as Figure 9, which includes 10 scatter subplots, where each subplot represents each method. From the figure, we can see the difference in the estimated and the measured PSD (of all size bins) for different methods. Some univariate methods show very poor results and now they are shown in the scatter plots. Some relevant explanations are also include in Section 4.1.